# On Compositional Uncertainty Quantification for Seq2seq Graph Parsing

**Zi Lin**[1,2]* **Du Phan**[2] **Panupong Pasupat**[2] **Jeremiah Liu**[2 3]† **Jingbo Shang**[1]†

[1]UC San Diego    [2]Google Research    [3]Harvard University

{lzi, jshang}@ucsd.edu    {phandu, ppasupat, jereliu}@google.com

## Abstract

Recent years have witnessed the success of applying seq2seq models to graph parsing tasks, where the outputs are compositionally structured (e.g., a graph or a tree). However, these seq2seq approaches pose a challenge in quantifying the model's compositional uncertainty on graph structures due to the gap between seq2seq output probability and structural probability on the graph. This work is the first to quantify and evaluate compositional uncertainty for seq2seq graph parsing tasks. First, we proposed a generic, probabilistically interpretable framework that allows correspondences between seq2seq output probability to structural probability on the graph. This framework serves as a powerful medium for quantifying a seq2seq model's compositional uncertainty on graph elements (i.e., nodes or edges). Second, to evaluate uncertainty quality in terms of calibration, we propose a novel metric called *Compositional Expected Calibration Error* (CECE) which can measure a model's calibration behavior in predicting graph structures. By a thorough evaluation for compositional uncertainty on three different tasks across ten domains, we demonstrate that CECE is a better reflection for distributional shift compared to vanilla sequence ECE. Finally, we validate the effectiveness of compositional uncertainty considering the task of collaborative semantic parsing, where the model is allowed to send limited subgraphs for human review. The results show that the collaborative performance based on uncertain subgraph selection consistently outperforms random subgraph selection (30% average error reduction rate) and performs comparably to oracle subgraph selection (only 0.33 difference in average prediction error), indicating that compositional uncertainty is an ideal signal for model errors and can benefit various downstream tasks. [1]

## 1 Introduction

Parsing a natural language sentence into a compositional graph structure, i.e., *graph parsing*, is an important task of natural language understanding beyond simple classification or text generation tasks. It has been broadly applied in applications like semantic parsing, code generation and knowledge graph generation. Recently, a line of research has successfully applied sequence-to-sequence (*seq2seq*) approaches to these graph parsing tasks (Vinyals et al., 2015; Xu et al., 2020; Orhan, 2021; Cui et al., 2022; Lin et al., 2022b). Despite achieving impressive results, these approaches pose a challenge in quantifying the model's predictive uncertainty on graph structures, making it hard to ensure a trustworthy and reliable deployment of NLP systems such as voice assistants (see an example in Figure 1). Meanwhile, most existing work on uncertainty estimation for seq2seq models focused on classification or language generation tasks (Kumar & Sarawagi, 2019; Vasudevan et al., 2019; Malinin & Gales, 2020; Jiang et al., 2021; Shelmanov et al., 2021; Wang et al., 2022; Pei et al., 2022). However, how to quantify and evaluate *compositional uncertainty*, the predictive uncertainty over compositional graph elements (i.e., nodes or edges), remains unresolved for seq2seq graph parsing (see related work in Appendix A.2). In this paper, we aim to answer these questions by proposing a simple probabilistic framework and rigorous evaluation metrics.

---

*Work done while interning at Google Research. †Co-senior authors.

[1]Open-source code may be found at https://github.com/google/uncertainty-baselines.

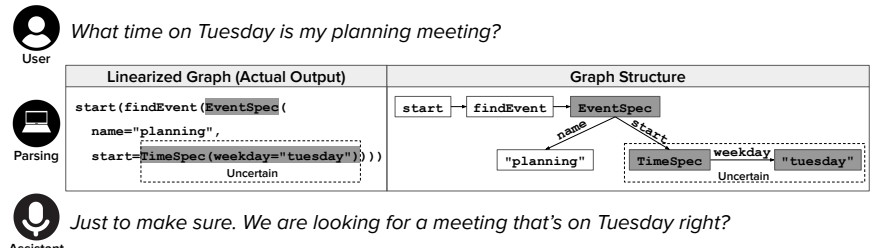

Figure 1: An example for graph semantic parsing in a dialogue system. The output of the model is a linearized semantic graph (left) that corresponds to the graph structure (right). The dotted square indicates the uncertain part, based on which the model can ask a clarification question.

Quantifying compositional uncertainty for seq2seq graph parsing is inherently more difficult than other seq2seq tasks like machine translation or speech recognition, since there is a gap between seq2seq output probabilities and conditional probabilities on the graph. Specifically, our interest is to express the conditional probability of a graph node $v$ concerning its parent $\mathrm{pa}(v)$, i.e., $p(v \mid \mathrm{pa}(v), x)$, rather than the likelihood of $v$ conditioning on the previous tokens in the linearized string. For example, in the graph structure of Figure 1, the subgraph rooted with node `TimeSpec` (in the dotted square) depends on its parent node `EventSpec`, while in the linearized graph, the parent node is not necessarily the previous token to the subgraph (the shaded spans). Consequently, we cannot directly quantify compositional uncertainty without bridging this gap between different probabilistic representations. To address this challenge, we propose a generic, probabilistic framework called *Graph Autoregressive Process* (GAP) (Section 2.1) that allows the correspondence between seq2seq output probability to graphical probability, i.e., assigning model probability for a node or edge on the graph. Thus, GAP can be used as a powerful medium for quantifying a seq2seq model's compositional uncertainty. Furthermore, to evaluate uncertainty quality, we propose a novel metric called *Compositional Expected Calibration Error* (CECE) to measure the model's behavior in predicting compositional graph structures (Section 2.2).

Taking semantic parsing as a canonical application, in Section 3, we build a large benchmark consisting of 3 semantic parsing tasks across 10 different domains, based on which we comprehensively evaluate compositional uncertainty under distributional shift and validate its effectiveness on a practical downstream task (collaborative semantic parsing).

First, in Section 3.1, we report different calibration metrics for a state-of-the-art seq2seq parser (Lin et al., 2022b) based on T5 (Raffel et al., 2020) as well as its different advanced uncertainty variants on the benchmark. We demonstrate that compared to vanilla ECE based on sequence accuracy, CECE is a better metric for reflecting distributional shift, i.e., task difficulty and domain generalization. We also notice that, despite the strong performance brought by those advanced uncertainty baselines on classification tasks, in the settings of graph parsing, the absolute advantage of these methods no longer holds when predicting graph edges. This suggests that developing uncertainty methods focused on compositional uncertainty can be a fruitful avenue for future research.

Second, in Section 3.2, we validate the practical effectiveness of compositional uncertainty considering the problem of collaborative semantic parsing. In this setting, the model is allowed to send a limited number of uncertain subgraphs for human review (see Figure 1 for an example). We test the collaborative performance on the benchmark, and find that using uncertain subgraph selection consistently outperforms random subgraph selection (selecting random subgraphs on the predicted graph) with an average error reduction rate of 30%, and performs fairly close to oracle subgraph selection (selecting incorrect subgraphs on the predicted graph) with an small difference in prediction error as 0.33. This indicates that compositional uncertainty is an ideal signal for the likelihood on model error over graph elements, and can benefit various downstream tasks, e.g., human-AI collaborative parsing and neural-symbolic parsing (Lin et al., 2022a).

In summary, our work makes the following contributions:

- **New Framework for Compositional Uncertainty Quantification.** We are the first to propose a simple and general probabilistic framework (GAP) that can quantify compositional uncertainty for seq2seq graph parsing. GAP allows us to go beyond the conventional autoregressive sequence

probability and express parent-child conditional probability on the graph, which is compatible with any graph parsing problem and autoresgressive model.

- **Rigorous Metrics for Compositional Uncertainty Calibration.** We introduce a novel quality measurement for compositional uncertainty (CECE) which can evaluate the model's calibration on graph elements and provide a better interpretation of the model's behavior in predicting graph structures under distributional shift.
- **Practical Effectiveness.** We comprehensively evaluate the calibration behavior of modern pre-trained large language models, i.e., T5 (Raffel et al., 2020), in compositional uncertainty quantification on a broad range of semantic parsing tasks of varying complexity (Redwoods, SNIPS, and SMCalFlow). We further evaluate the benefit of compositional uncertainty quantification in enabling new capacity (e.g, fine-grained collaborative prediction for complex semantic parsing) in downstream applications. Specifically, our results show that compositional uncertainty can significantly benefit collaborative parsing performance, with only a 0.33 difference in prediction error compared to the headroom.

## 2  QUANTIFYING AND EVALUATING COMPOSITIONAL UNCERTAINTY

**Problem Formulation.** In this work, we interpret the term *graph parsing* as mapping from surface strings (usually natural language sentences) to target representations that are explicitly or implicitly graph-structured. Formally, the input is a natural language utterance $x$, and the output is a DAG $G = \langle \mathbf{N}, \mathbf{E} \rangle$, where $\mathbf{N}$ is the set of nodes and $\mathbf{E} \in \mathbf{N} \times \mathbf{N}$ is the set of edges. In the case of seq2seq parsing, $G$ is represented as a linearized graph string $g = s_1 s_2 \cdots s_L$ consists of symbols $\{s_l\}_{l=1}^{L}$. In the experiment, we use PENMAN notation (Kasper, 1989) to linearize all the formalism, which is a serialization format for the directed, rooted graphs used to encode semantic dependencies (more details are available in Appendix B).

To this end, our goal is to quantify the graph-level uncertainty $p(G|x)$ from the sequence-level probability $p(g|x)$ generated by a seq2seq model, which we term the *compositional uncertainty*. For example, our interest to express the conditional probability of a graph node $v$ with respect to its parent $\mathrm{pa}(v)$, i.e., $p(v|\mathrm{pa}(v), x)$, rather than the likelihood of $v$ conditioning on the previous tokens in the linearized string. In the following sections, we will introduce how to quantify (Section 2.1) and evaluate (Section 2.2) this compositional uncertainty.

### 2.1  QUANTIFYING COMPOSITIONAL UNCERTAINTY VIA GRAPH AUTOREGRESSIVE PROCESS (GAP)

To properly model the uncertainty $p(G|x)$ from a seq2seq model, we need an intermediate probabilistic representation to translate the raw token-level probability to the distribution over graph elements (i.e., nodes and edges). To this end, we introduce a simple probabilistic formalism termed *Graph Autoregressive Process* (GAP), which is a probability distribution assigning seq2seq learned probability to the graph elements $v \in G$.

Specifically, as the seq2seq-predicted graph adopts both a sequence-based representation $g = s_1, ..., s_L$ and a graph representation $G = \langle \mathbf{N}, \mathbf{E} \rangle$, the GAP model adopts both an autoregressive representation $p(g|x) = \prod_i p(s_i|s_{<i}, x)$ analogous to that of the seq2seq model (Section 2.1.1), and also a probabilistic graphical model representation $p(G|x) = \prod_{v \in G} p(v|\mathrm{pa}(v), x)$ for proper quantification of model uncertainty on the graph (Section 2.1.2). Both representations share the same set of underlying probability measures (i.e., the graphical-model likelihood $p(G|x)$ can be derived from the autoregressive probabilities $p(s_i|s_{<i}, x)$). As we will show, GAP serves as a powerful medium for quantifying compositional uncertainty for seq2seq graph parsing.

### 2.1.1  AUTOREGRESSIVE REPRESENTATION FOR LINEARIZED SEQUENCE $g$

Given an input sequence $x$ and output sequence $y = y_1 y_2 \cdots y_N$, the token-level autoregressive distribution from a seq2seq model is $p(y|x) = \prod_{i=1}^{N} p(y_i|y_{<i}, x)$. In the context of graph parsing, the output sequence describes a linearized graph $g = s_1 s_2 \cdots s_L$, where each symbol $s_i = \{y_{i_1} y_{i_2} \cdots y_{i_{N_i}}\}$ represents either a node $n \in \mathbf{N}$ or an edge $e \in \mathbf{E}$ of the graph and corresponds to a collection of beam-decoded tokens $\{y_{i_1} y_{i_2} \cdots y_{i_{N_i}}\}$. This process is illustrated as follows:

To this end, GAP assigns probability to each linearized graph $g = s_1 s_2 \cdots s_L$ autoregressively as $p(g|x) = \prod_{i=1}^{L} p(s_i|s_{<i}, x)$, and the conditional probability $p(s_i|s_{<i}, x)$ is computed by aggregating the token probability:

$$p(s_i|s_{<i}, x) = p(\{y_{i_1} \cdots y_{i_{N_i}}\}|s_{<i}, x) = \prod_{j=1}^{N_i} p(y_{i_j}|y_{i_{<j}}, s_{<i}, x) \tag{1}$$

**Marginal and Conditional Probability.** Importantly, GAP allows us to compute the marginal and (non-local) conditional probabilities for graph elements $s_i$. Given the input $x$, the marginal probability of $s_i$ is computed as

$$p(s_i|x) = \int_{s_{<i}} p(s_i|s_{<i}, x)p(s_{<i}|x)\mathrm{d}s_{<i} \tag{2}$$

by integrating over the space of all possible subsequences $s_{<i}$ before the symbol $s_i$. Then, the (non-local) conditional probability between two graph elements $(s_i, s_j)$ with $i < j$ is computed as

$$p(s_j|s_i, x) = \int_{s_{i \to j}, s_{<i}} p(s_j, s_{i \to j}|s_i, s_{<i}, x)p(s_i|s_{<i}, x)p(s_{<i}|x)\mathrm{d}s_{i \to j}\mathrm{d}s_{<i} \tag{3}$$

by integrating over the space of subsequences $s_{i \to j}$ between $(s_i, s_j)$ and the subsequence $s_{<i}$ before $s_i$. Higher order conditional (e.g., $p(s_j|(s_i, s_l), x)$) can be computed analogously. Notice this gives us the ability to reason about long-range dependencies between non-adjacent symbols on the sequence. Furthermore, the conditional probability on the *reverse* direction can also be computed using the Bayes' rule: $p(s_i|s_j, x) = \frac{p(s_j|s_i, x)p(s_i|x)}{p(s_j|x)}$.

**Efficient Estimation Using Beam Outputs.** In practice, we can estimate $p(s_i|x)$ and $p(s_j|s_i, x)$ efficiently via importance sampling using the output from the beam decoding $\{g_k\}_{k=1}^{K}$, where $K$ is the beam size (Malinin & Gales, 2020). The marginal probability can be computed as

$$\hat{p}(s_i|x) = \sum_{k=1}^{K} p(s_i|s_{k,<i}, x) * \pi_k, \pi_k = \frac{\exp(\frac{1}{t} \log p(g_k|x))}{\sum_{k=1}^{K} \exp(\frac{1}{t} \log p(g_k|x))} \tag{4}$$

here $\pi_k$ is the importance weight proportional to the beam candidate $g_k$'s log likelihoods, and $t > 0$ is the temperature parameter fixed to a small constant (e.g., $t = 0.1$) (Malinin & Gales, 2020). If the symbol $s_i$ does not appear in the $k^{\text{th}}$ beam, we set $p(s_i|s_{k,<i}, x) = 0$. As shown, the marginalized probability $\hat{p}(s_i|x)$ provides a way to reason about the *global* importance of $s_i$ by integrating the probabilistic evidence $p(s_i|s_{k,<i}, x)$ over the whole beam-sampled posterior space. It is able to capture the cases of spurious graph elements $s_i$ with high local probability $p(s_i|s_{k,<i}, x)$ but low global likelihood (i.e., only appear in a few low-probability beam candidates). Therefore, it is a useful quantity for structure induction (e.g., edge and node pruning) in graphical model inference (Dianati, 2016).

Then, for two symbols $(s_i, s_j)$ with $i < j$, we can estimate the conditional probability as

$$\hat{p}(s_j|s_i, x) = \sum_{k=1}^{K} p(s_j|s_i, s_{k,i \to j}, s_{k,<i}, x) * \pi_k^i, \pi_k^i = \frac{\exp(\frac{1}{t} \log p(g_k|x)) * I(s_i \in g_k)}{\sum_{k=1}^{K} \exp(\frac{1}{t} \log p(g_k|x)) * I(s_i \in g_k)} \tag{5}$$

here $\pi_k^i$ is the importance weight among beam candidates that contains $s_i$. Notice this is different from Equation 4 where $\pi_k$ is computed over all beam candidates regardless of whether it contains $s_i$.

### 2.1.2 PROBABILISTIC GRAPHICAL MODEL REPRESENTATION FOR $G$

So far, we have focused on probability computation based on the graph's linearized representation $p(g|x) = \prod_i p(s_i|s_{<i}, x)$. In this section, we further consider GAP's graphical model representation $p(G|x) = \prod_{v \in G} p(v|\operatorname{pa}(v), x)$.

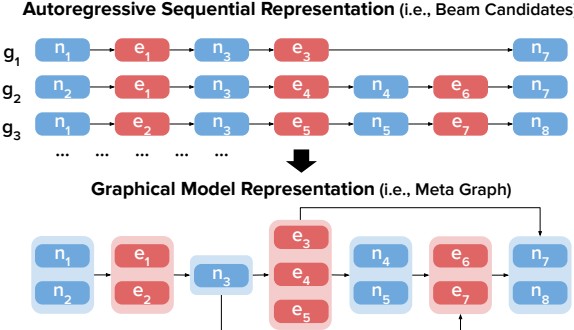

Figure 2: Visual illustration of constructing graphical model representation for meta graph $\mathcal{G}$ from autoregressive representation $\{g_k\}_{k=1}^K$.

Specifically, given a beam sample of linearized graphs $\{g_k\}_{k=1}^K$, well-established algorithms exist to synthesize different graph predictions into a meta graph $\mathcal{G}$. Briefly, these methods first convert each string $g_b$ into their original graph representation $G_k = \langle \mathbf{N}_k, \mathbf{E}_k \rangle$, then merge multiple graphs $\{G_k\}_{k=1}^K$ using a graph matching algorithm (Cai & Knight, 2013; Hoang et al., 2021).

A visual illustration of the resulting graph $\mathcal{G}$ is shown in Figure 2, where $n_i$ and $e_j$ are the candidates for the node and edge predictions collected from beam sequences. As shown, compared to the sequence-based representation $g$, the meta graph $\mathcal{G}$ (1) explicitly enumerates different candidates for each node and edge prediction (e.g., $n_1$ v.s. $n_2$ for predicting the first element), and (2) provides an explicit account of the parent-child relationships between variables on the graph (e.g., $e_7$ is a child node of $n_3$ in the predicted graph, which is not reflected in the autoregressive representation).

From the probabilistic learning perspective, the meta graph $\mathcal{G}$ describes the space of possible graphs (i.e., the *support*) for a graph distribution $p(G|x) : G \to [0, 1]$. It describes the possible node and edge variables and their dependencies on the graph $G$ (i.e., the shaded squares in the Figure 2), and also different possible values for each node and edge variable (i.e., the solid squares within each shaded square in Figure 2).

To this end, GAP assigns proper graph-level probability $p(G|x)$ to graphs $G$ sampled from the meta graph $\mathcal{G}$ via the graphical model likelihood:

$$p(G|x) = \prod_{v \in G} p(v|\operatorname{pa}(v), x) = \prod_{n \in \mathbf{N}} p(n|\operatorname{pa}(n), x) * \prod_{e \in \mathbf{E}} p(e|\operatorname{pa}(e), x) \tag{6}$$

where $p(v|\operatorname{pa}(v), x)$ is the conditional probability for $v$ with respect to their parents $\operatorname{pa}(v)$ in $G$. Given the candidates graphs $\{G_k\}_{k=1}^K$, we can express the likelihood for $p(v|\operatorname{pa}(v), x)$ by writing down a multinomial likelihood enumerating over different values of $\operatorname{pa}(v)$ (Murphy, 2012).

This in fact leads to a simple expression for the model likelihood as a simple averaging of the beam-sequence log likelihoods:

$$\log p(n|\operatorname{pa}(n), x) \propto \frac{1}{K} \sum_{k=1}^K \log p(n|\operatorname{pa}(n) = c_k) \tag{7}$$

where $c_k$ is the value of $\operatorname{pa}(n)$ in $k^{\text{th}}$ beam sequence, and the conditional probabilities are computed using Equation (5). See Appendix C for a detailed derivation.

In summary, for each graph element variable $v \in G$, GAP allows us to compute the graphical-model conditional likelihood $p(v|pa(v), x)$ via its graphical model representation, and also to compute the marginal probability $p(v|x)$ via its autoregressive presentation. Algorithm 1 summarizes the full GAP computation.

## 2.2 Evaluating Compositional Uncertainty

In this section, we present how to evaluate compositional uncertainty. A common approach to evaluate a model's uncertainty quality is to measure its calibration performance, i.e., whether the

---

**Algorithm 1** Graph Autoregressive Process (GAP)

---

**Inputs:**
 Beam candidates with probabilities $\{p(g_k|x)\}_{k=1}^{K}$, *Meta graph* $\mathcal{G}$
**Output:**
 Marginal probabilities $\{p(s|x)\}$, Graph model likelihood $\log p(G|x)$
**for** $v \in \mathcal{G}$ **do**
 Compute marginal likelihood:  $p(v = s|x)$ (Equation 4)
 Compute graphical model likelihood:  $\log p(v = s|\operatorname{pa}(v), x)$ (Equation 7)
**Return**
 Marginal Probabilities: $\{p(v|x)\}$
 Graphical Model Likelihood: $\log p(G|x)) = \sum_{v \in G} \log p(v|\operatorname{pa}(v), x)$

---

model's predictive uncertainty is indicative of the predictive error, e.g., expected calibration error (ECE; Naeini et al., 2015). In this work, we propose a compositional calibration metric based on ECE, which measures the difference in expectation between the model's predictive confidence (e.g., the maximum probability score) on graph elements (nodes or edges) and their actual match to the gold graph.

Formally, during inference time, given a input $x$ and a target graph $G$, we first partition the confidence interval into $B$ equal bins $I_1, \ldots, I_B$. Then in each measure the absolute difference between the node/edge accuracy and confidence of predictions in that bin. This gives the compositional expected calibration error ($\text{CECE}_G$) for graph $G$ as:

$$\frac{1}{|G|} \sum_{b=1}^{B} \left| \sum_{\hat{v}_t \in \hat{G}, p(\hat{v}_t|x) \in I_b} C(\hat{v}_t, G) - p(\hat{v}_t|x) \right| \tag{8}$$

where $|G|$ is the number of graph elements in the target graph $G$, $\hat{v}_t$ is the $t^{\text{th}}$ element (node/edge) in the predicted graph $\hat{G}$, $C(\hat{v}_{it}, G_i)$ denotes if $\hat{v}_{it}$ matches in the graph $G_i$ using a graph matching algorithm, and $p(\hat{v}_t|x)$ can be obtained by GAP in Section 2.1. Specifically, we use the matching algorithm adopted in SMATCH (Cai & Knight, 2013), which is the same graph matching algorithm for constructing *meta graph* $\mathcal{G}$ (see Appendix E for details). Alternatively, we can compute CECE only for node/edge predictions, namely $\text{CECE}_{\mathbf{N}}$ and $\text{CECE}_{\mathbf{E}}$.

## 3 EXPERIMENTAL EVALUATION

**Datasets.** In this paper, we take semantic parsing as a canonical application, and build a large benchmark consisting of three semantic parsing tasks and covering ten different domains, ranging from graph-based grammar parsing to dialogue-oriented semantic parsing:

- **Redwoods:** The LinGO Redwoods Treebank is a collection of hand-annotated corpora for an English grammar consisting of more than 20 datasets. The underlying grammar is called English Resource Grammar (ERG; Flickinger et al., 2014; Bender et al., 2015), which is an open-source, domain-independent, linguistically precise, and broad-coverage grammar. ERG can be presented into different types of annotation formalism. This work focuses on the Elementary Dependency Structure (EDS; Oepen & Lønning, 2006), which is a compact representation that can be expressed as a DAG. Following previous works, for in-domain test, we train and evaluate models on the subset treebank corresponding to the 25 Wall Street Journal (WSJ) sections with standard data splits (Flickinger et al., 2012). For out-of-domain (OOD) evaluations, we select 7 diverse datasets from Redwoods: Wikipedia (Wiki), the Brown Corpus (Brown), the Eric Raymond Essay (Essay), customer emails (E-commerce), meeting/hotel scheduling (Verbmobil), Norwegian tourism (LOGON) and the Tanaka Corpus (Tanaka) (See Appendix D for more details).
- **SMCalFlow:** SMCalFlow (Andreas et al., 2020) is a large corpus of semantically detailed annotations of task-oriented natural dialogues. The annotation uses dataflow computational graphs, composed of a rich set of both general and application specific functions, to represent user requests as rich compositional expressions. We use the standard data split in the original paper and evaluate inference results on development set.
- **SNIPS:** SNIPS (Coucke et al., 2018) is a slot filling dataset containing 39 slot names from 7 different domains collecting from the SNIPS voice assistant. It is usually formulated as a sequence

| | Dataset | Performance Metrics | | | | Calibration Metrics | | | | |
|---|---|---|---|---|---|---|---|---|---|---|
| | | $\text{ACC}_{\text{seq}}(\uparrow)$ | $\textsc{Smatch}(\uparrow)$ | $\text{F1}_{\mathbf{N}}(\uparrow)$ | $\text{F1}_{\mathbf{E}}(\uparrow)$ | $\text{ECE}_{\text{seq}}(\downarrow)$ | $\text{CECE}_G(\downarrow)$ | $\text{CECE}_{\mathbf{N}}(\downarrow)$ | $\text{CECE}_{\mathbf{E}}(\downarrow)$ | |
| ↑ | WSJ (In-domain) | 51.01 | 95.92 | 97.38 | 94.35 | 0.3632 | 0.0345 | 0.0221 | 0.0465 | Min |
| Domain distance | Tanaka | 68.10 | 95.27 | 95.65 | 94.95 | 0.2562 | 0.0452 | 0.0405 | 0.0498 | |
| | E-commerce | 51.53 | 93.48 | 94.18 | 92.88 | 0.3840 | 0.0721 | 0.0652 | 0.0793 | |
| | Brown | 40.93 | 92.32 | 93.81 | 90.89 | 0.3969 | 0.0673 | 0.0530 | 0.0824 | |
| | Essay | 30.80 | 92.07 | 94.18 | 90.96 | 0.4589 | 0.0691 | 0.0565 | 0.0808 | |
| | LOGON | 31.45 | 91.10 | 92.58 | 90.70 | 0.5148 | 0.0834 | 0.0726 | 0.0944 | |
| | Verbmobil | 51.45 | 90.38 | 91.45 | 90.02 | 0.3298 | 0.0960 | 0.0861 | 0.1063 | |
| ↓ | Wiki | 28.16 | 89.64 | 90.65 | 89.37 | 0.4105 | 0.1038 | 0.0939 | 0.1135 | |
| | SMCalflow | 82.83 | 93.87 | 94.23 | 93.48 | 0.1427 | 0.0651 | 0.0605 | 0.0702 | |
| | SNIPS | 91.14 | - | 98.49 | - | 0.0756 | - | 0.0156 | - | Max |

Table 1: Evelution results on three tasks. $\text{Acc}_{\text{seq}}$ means sequence accuracy (exact match); $\text{F1}_{\mathbf{N}}$/$\text{F1}_{\mathbf{E}}$ means F1 scores for nodes/edges; $\text{CECE}_G$/$\text{CECE}_{\mathbf{N}}$/$\text{CECE}_{\mathbf{E}}$ means compositional ECE for graph/nodes/edges. Since we use pseudo edges to transfer examples to graphs for SNIPS, we skip edge-related evaluations for SNIPS. The background color in calibration metrics indicates the number order in column (Green: Min; Red: Max).

labeling problem. Following previous work (Yu et al., 2021), we train models on five source domains, use a sixth one for development, and test on the remaining domian.

We transfer all the data into linearized graphs (Appendix B). To reduce the sequence length, since the number of node/edge names are limited, we set them untokenizable to the tokenizer.

**Evaluation Metrics.** Consistent with previous work, the performance metric used for Redwoods is $\textsc{Smatch}$ score (Cai & Knight, 2013), which computes the degree of overlap between two semantic graphs (see Appendix E for details). For SMCalFlow, we use sequence accuracy (exact match). For SNIPS, we use slot F1 score, which is equal to node F1 score when we transfer the SNIPS data into PENMAN notation. For model calibration, we report naive ECE based on sequence accuracy and compositional uncertainty CECE introduced in Section 2.2.

**Seq2seq Model.** We adopt T5 (Raffel et al., 2020) as the baseline model, which is a pre-trained seq2seq Transformer model that has been widely used in many NLP applications. We use the open-sourced T5X, which is a new and improved implementation of T5 codebase in JAX and Flax. Specifically, we use the official pretrained T5-large (770 million parameters) and finetuned it on three datasets respectively. By evaluating the performance metrics for each task, we find that the T5 model is capable of achieving the state-of-the-art results on all the tasks compared to previous work (see Table 3 in Appendix F for full comparison).

## 3.1 Evaluating Compositional Uncertainty under Distributional Shift

**Comparing Compositional ECE to vanilla ECE.** Table 1 reports the evaluation results on the benchmark. First, we find that sequence accuracy ($\text{ACC}_{\text{seq}}$) does not necessarily correlate to the $\textsc{Smatch}$ score, which makes the vanilla ECE based on sequence accuracy, i.e., $\text{ECE}_{\text{seq}}$, less informative in reflecting model's calibration on predicting graph structures. Second, comparing to Redwoods (in-domain) which requires parsing a natural language sentence into a pre-defined grammar representation, and SNIPS which requires labeling intent slots for a natural language sentence, SMCalFlow is more ambiguous and difficult as it involves complex dialogue histories and fine-grained intent slots (Andreas et al., 2020). We notice that the CECE is larger for SMCalFlow on in-domain test, indicating that CECE is a better metrics to reflect this task ambiguity/difficulty. Finally, as the model generalization degrades across different domains for Redwoods, CECE also increases accordingly, indicating that CECE can reflect model's generalization under domain shift.

**Comparing Advanced Uncertainty Methods.** In recent year, a variety of methods have been developed to improve the DNNs uncertainty quality for classification problems. Here, we are interested to understand if the benefit brought by those advanced methods to classification setting translates also to the graph parsing setting. In this section, we evaluate the performance of 6 uncertainty baselines on Redwoods across 8 different domains. We consider T5-Large (Raffel et al., 2020) as the base model, and select six methods based on their practical applicability for the base model.

Specifically, we consider (1) *Deterministic* model which is the base T5 model; (2) *Monte Carlo Dropout* (*MC Dropout*) which estimates uncertainty using the Monte Carlo average of 5 dropout samples (Gal & Ghahramani, 2016); (3) *Deep Ensemble* (*DE*) which trains 4 deterministic models individually and averages all predictions (Lakshminarayanan et al., 2017); (4) *Batch Ensemble* (*BE*),

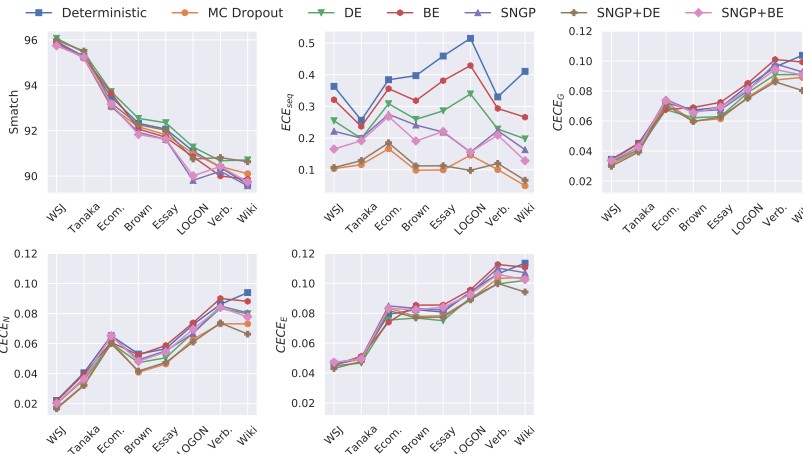

Figure 3: Evaluation results for different uncertainty baselines in terms of SMATCH score, ECE$_{\text{seq}}$, CECE$_G$, CECE$_{\mathbf{N}}$, and CECE$_{\mathbf{E}}$ under distributional shift.

an ensemble method which has much lower computational and memory costs comparing to MC Dropout and Deep Ensemble (Wen et al., 2019); (5) *Spectral-normalized Neural Gaussian Process* (*SNGP*), a recent state-of-the-art approach which improves uncertainty quality by transforming a neural network into an approximate Gaussian process model (Liu et al., 2020); (6) *SNGP+DE* which is the deep ensemble for 4 individual SNGP models; (7) *SNGP+BE* which uses a combination of Batch Ensemble and SNGP layers. The results are shown in Figure 3. The full evaluation results can be found in Appendix G (Table 4).

**Results.** As shown in the figure, these uncertainty baselines generally follows the same pattern on domain shift, i.e., decrease in SMATCH corresponds to increase in CECE$_G$, while we cannot infer distributional shift via ECE$_{\text{seq}}$. Some uncertainty baselines (e.g., MC Dropout and SNGP+DE) can achieve better in both ECE$_{\text{seq}}$ and CECE$_G$ compared to deterministic model across different domains, where MC Dropout achieves the best results in ECE$_{\text{seq}}$ and SNGP+DE achieves the best results in CECE$_G$. By further evaluating CECE$_{\mathbf{N}}$ and CECE$_{\mathbf{E}}$, we notice that the improvement in CECE$_G$ mainly comes from node predictions (the difference in CECE$_{\mathbf{N}}$ is more obvious than CECE$_{\mathbf{E}}$), while for edge predictions, only little improvement is observed. This suggests that uncertainty estimation is structurally different for seq2seq graph parsing tasks compared to classification tasks, and further research is needed for designing better calibrated model with more focus on compositional uncertainty calibration.

### 3.2 PRACTICAL EFFECTIVENESS: UNCERTAINTY-GUIDED COLLABORATIVE SEMANTIC PARSING

**Motivation.** To further explore the correlation between model uncertainty and performance, we plot the histogram for the T5 model's probabilities verses the node/edge accuracies in Appendix H (Figure 5), where we find that low model probability generally corresponds to low model performance. This serves as a motivation for collaborative semantic parsing using composotional uncertainty, where the model is allowed to send a limited number of uncertain subgraphs for human review. This is a practical setting in lots of realistic scenarios, for example, in Figure 1, the model can ask for clarification regarding the uncertain subgraph (dotted squared) via modeling the uncertainty score for each element in the parsed semantic graph. This process allows the system to collaborate with the users to avoid triggering unwanted actions, which cannot be achieved without properly quantifying compositional uncertainty over graph elements.

**Uncertainty-based Subgraph Selection.** For a well-trained seq2seq parser in Section 3.1, we find uncertain subgraphs by (1) finding $e$ uncertain nodes as root nodes based on ranked compositional uncertainty scores over the predicted graph; (2) tracing descendants from root nodes up to depth $d$.[2]

---

[2]Here the number of subgraphs $e$ and the maximum depth $d$ are our review capacity for uncertain subgraphs (in experiment, we try $e \in [1, 3, 5]$ and $d \in [1, 2, 3]$).

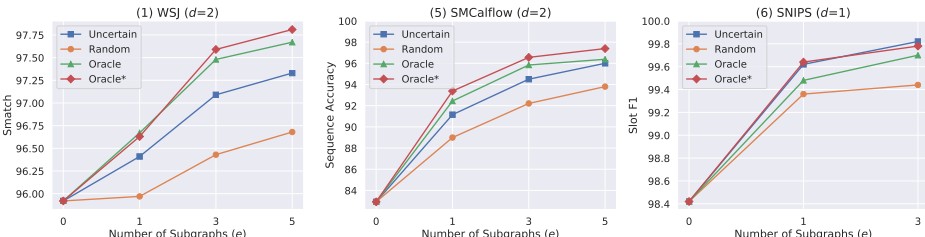

Figure 4: Collaborative performance for different semantic parsing tasks. The performance metrics are SMATCH score for Redwoods, sequence accuracy for SMCalflow and slot (node) F1 for SNIPS

**Baselines for Comparision.** We take another three subgraph selections as comparision: (1) *Random* subgraph selection by randomly finding $e$ nodes as root nodes and tracing descendants from roots up to depth $d$; (2) *Oracle* subgraph selection by finding $e$ incorrectly predicted nodes as root nodes and tracing descendants from roots up to depth $d$; (3) *Oracle\** subgraph selection by finding $e$ incorrectly predicted nodes (prioritizing the most uncertain nodes) and tracing descendants from roots up to depth $d$.

**Training and Inference.** The training examples for the collaborative model are generated by attaching human edit results of random subgraphs to input sentences [3]. During the inference, the test examples are generated by attaching corresponding human edit results for each subgraph selection strategy to input sentences.

**Results.** The results are shown in Figure 4. Due to space limitation, we only report results on $d = 2$ for Redwoods and SMCalflow, and $d = 1$ for SNIPS. Full results for different combinations of example numbers and depths are reported in Table 5 and Table 6 (Appendix I). We see that for all three tasks, uncertainty-based subgraph selection consistently outperforms random subgraph selection with average error reduction rates 13.64%, 24.45% and 52.11% respectively, and performs fairly close to *Oracle* with an average difference in prediction error as small as 0.33. This shows that compositional uncertainty is effective in detecting potential incorrect subgraph predicted by the model. Meanwhile, we notice that *Oracle\** performs better than *Oracle*, indicating that incorrect predicted subgraphs with high uncertainty are more informative to the collaborative model.

**Analysis.** Theoretically, the performance of collaborative parsing is determined by how many incorrectly predicted subgraphs can be selected for human edits, where *Oracle* is the headroom given limited budgets. In Table 7 (Appendix I), We further conduct an analysis to subgraphs selected by different strategies by calculating the coverage rate of incorrect nodes in subgraphs to incorrect nodes in the entire graph (i.e., error node coverage rate). The results indicates that compositional uncertainty is effective in detecting incorrectly predicted nodes.

## 4 CONCLUSION AND FUTURE WORK

Over the past few years, lots of efforts have been made to apply seq2seq models to graph parsing, which is an important area in NLP. Despite achieving the state-of-the-art in various graph parsing tasks, these seq2seq approaches pose a challenge on how to interpret model's predictive uncertainty on predicting graph structures. This work is the first to provide a general method to properly quantify and evaluate compositional uncertainty for seq2seq graph parsing, which is achieved by a simple probabilistic framework (GAP) and a rigorous metric (CECE). The experimental evaluation demonstrates that CECE is an effective metric to reflect distributional shift and compositional uncertainty is a useful tool for downstream tasks such as collaborative semantic parsing. For future work, several directions are worth investigations, including: (1) evaluating the practical effectiveness of advanced uncertainty methods in collaborative semantic parsing; (2) learning theoretic experiments to interrogate the seq2seq model's ability in capturing the true distribution of the probabilistic graphical model; (3) extending the benefit of compositional uncertainty to more real-world NLP tasks, e.g., retrieval-augmented semantic parsing (Hashimoto et al., 2018) and active learning.

---

[3]The model is of the same setting as in Section 3.1. Here we take gold subgraphs as human edit results.

ACKNOWLEDGMENTS

We appreciate the insightful comments from the anonymous reviewers. We would like to thank Jie Ren for discussion and proofreading.

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

## A   RELATED WORK

### A.1   SEQ2SEQ GRAPH PARSING

Seq2seq graph parsing is inspired by the success of recent seq2seq models (particularly pretrained models), which are the heart of modern neural machine translation. This type of parser encodes and views a target graph as a string from another language (Vinyals et al., 2015).

However, simply applying seq2seq models to graph parsing is not always successful when the target graph is complicated, e.g., for Abstract Meaning Representation (AMR; Banarescu et al., 2013) or English Resource Grammar (ERG; Flickinger et al., 2014). This is because effective linearization (encoding graphs as linear sequences) and data sparsity were thought to pose significant challenges (Konstas et al., 2017). Alternatively, some specifically designed preprocessing procedures for vocabulary and entities can help to address these issues (Konstas et al., 2017; Peng et al., 2017). These preprocessing procedures are very specific to a certain type of meaning representation and are difficult to transfer to others. To address this, (Bevilacqua et al., 2021) propose to use special tokens to represent variables in the linearized graph and to handle co-referring nodes. Lin et al. (2022b) propose a variable-free top-down linearization and a compositionality-aware tokenization for ERG graph preprocessing, and successfully transfer the ERG parsing into a translation problem that can be solved by a state-of-the-art seq2seq model T5 (Raffel et al., 2020). Their parser achieves the best known results on the in-domain test set for ERG parsing.

### A.2   UNCERTAINTY QUANTIFICATION FOR GRAPH PARSING

Compared to seq2seq graph parsing, uncertainty quantification is straightforward if the parser explicitly models the target graph structures, e.g., chart parsers (Magerrnan & Marcus, 1991), factorization-based parsers (McDonald, 2006; Cao et al., 2021) or composition-based parsers (Chen et al., 2018; 2019), given that the model's score function is naturally aligned with the graph structure. As for transition-based parser (Fernandez Astudillo et al., 2020; Zhou et al., 2021), where the target graph is generated via a series of actions, in a process that is very similar to dependency tree parsing (Yamada & Matsumoto, 2003; Nivre, 2008), some previous work has used important sampling to estimate probabilities (Dyer et al., 2016), and model uncertainty for alignments between graph nodes and input text tokens (Drozdov et al., 2022). These works are very specific to the formalism of the target graph, and it is difficult to transfer to other graph parsing problems.

Some previous uncertainty quantification methods have focused on sequential or token-level uncertainty for seq2seq model. For example, Dong et al. (2018) model uncertainty for neural semantic parsers by outlining three major causes of uncertainty including model uncertainty, data uncertainty and input uncertainty, and design various metrics to quantify these factors. Lin et al. (2022b) uses predictive probability generated by the T5 model as a signal for neural-symbolic parsing. However, these uncertainty quantification methods cannot model compositional uncertainty over the graph structures.

## B   PENMAN NOTATION

PENMAN notation, originally called Sentence Plan Notation in the PENMAN project (Kasper, 1989), is a serialization format for the directed, rooted graphs used to encode semantic dependencies,

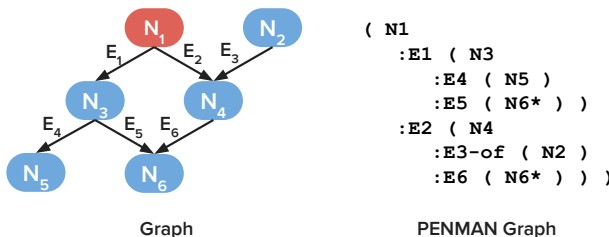

```
( N1
   :E1 ( N3
      :E4 ( N5 )
      :E5 ( N6* ) )
   :E2 ( N4
      :E3-of ( N2 )
      :E6 ( N6* ) ) )
```

Graph                               PENMAN Graph

most notably in the Abstract Meaning Representation (AMR) framework (Banarescu et al., 2013). It looks similar to Lisp's S-Expression in using parentheses to indicate nested structures. To make PENMAN notation compatible with the seq2seq learning, we adopted a variable-free version of PENMAN which was first proposed in Lin et al. (2022b). The general template is illustrated as follows:

The linearized form can only describe projective structures such as trees, so in order to capture non-projective graphs, this notation (1) reverse some of the edges to make it can be written in top-down tree order, e.g., `:E3-of` here (2) use star markers to indicate a node referred later to establish a reentrancy, e.g., `E6*` here. Table 2 shows some variable-free PENMAN linearized examples for different semantic parsing tasks.

| Datasets | Inputs | Outputs |
|---|---|---|
| Redwoods | *The Pentagon foiled the plan.* | `( _foil_v_1`
`  :ARG1 ( named :carg "Pentagon"`
`    :BV-of ( _the_q ) )`
`  :ARG2 ( _plan_n_1`
`    :BV-of ( _the_q ) ) )` |
| SMCalflow | *User: What time on Tuesday is my planning meeting?* | `( start`
`  :ARG1 ( findEvent`
`    :ARG1 ( EventSpec :name "planning"`
`      :start ( Timespec :weekday "tuesday" ) ) ) )` |
| SNIPS | *Find a movie called Living in America.* | `( IN:SEARCH_CREATIVE_WORK`
`   :ARG1 ( SL:OBJECT_TYPE :carg "movie" )`
`   :ARG2 ( SL:OBJECT_NAME :carg "living_in_america" ) )` |

Table 2: Examples for variable-free PENMAN linearized graph (template can be found in Appendix B) in three different semantic parsing tasks (task details can be found in Section 3). Here `:carg` means corresponding spans in the sentence.

## C  SIMPLIFIED EXPRESSION FOR GRAPHICAL MODEL LIKELIHOOD

Given the candidates graphs $\{G_k\}_{k=1}^{K}$, we can express the likelihood for $p(v|\operatorname{pa}(v), x)$ by writing down a multinomial likelihood enumerating over different values of $\operatorname{pa}(v)$ (Murphy, 2012). For example, say $\operatorname{pa}(n) = (e_1, e_2)$ which represents a subgraph of two edges $(e_1, e_2)$ pointing into a node $n$. Then the conditional probability $p(n|\operatorname{pa}(n), x)$ can be computed by enumerating over the observed values of $(e_1, e_2)$ pair:

$$p(n|\operatorname{pa}(n), x) = p(n|(e_1, e_2), x) \propto \prod_{c \in \text{Candidate}(e_1, e_2)} p(n|(e_1, e_2) = c, x)^{K_c} \qquad (9)$$

where Candidate$(e)$ is the collection of possible symbols $s$ the variable $e$ can take, and $K_c$ is the number of times $(e_1, e_2)$ takes a particular value $c \in \text{Candidate}(e_1, e_2) = \text{Candidate}(e_1) \times \text{Candidate}(e_2)$.

Then, the log likelihood becomes:

$$\log p(n|\operatorname{pa}(n), x) = \sum_c K_c * \log p(n|(e_1, e_2) = c) \qquad (10)$$

To simplify this above expression, we notice that $\log p(n|\operatorname{pa}(n), x)$ can be divided by the constant beam size K without impacting the inference. As a result, the log probability can be computed by

simplify averaging the values of $\log p(v|\operatorname{pa}(v) = c_k)$ across the beam candidates:

$$\log p(n|\operatorname{pa}(n), x) \propto \sum_c \frac{K_c}{K} \log p(n|(e_1, e_2) = c) = \frac{1}{K} \sum_{k=1}^{K} \log p(n|(e_1, e_2) = c_k) \tag{11}$$

where $c_k$ is the value of $(e_1, e_2)$ in $k^{\text{th}}$ beam candidate.

## D OOD DATASETS FOR ERG PARSING

**Wikipedia (Wiki)**  The DeepBank team constructed a treebank for 100 Wikipedia articles on Computational Linguistics and closely related topics. The treebank of 11,558 sentences comprises 16 sets of articles. The corpus contains mostly declarative, relatively long sentences, along with some fragments.

**The Brown Corpus (Brown)**  The Brown Corpus was a carefully compiled selection of current American English, totalling about a million words drawn from a wide variety of sources.

**The Eric Raymond Essay (Essay)**  The treebank is based on translations of the essay "The Cathedral and the Bazaar" by Eric Raymond. The average length and the linguistic complexity of these sentences is markedly higher than the other treebanked corpora.

**E-commerce**  While the ERG was being used in a commercial software product developed by the YY Software Corporation for automated response to customer emails, a corpus of training and test data was constructed and made freely available, consisting of email messages composed by people pretending to be customers of a fictional consumer products online store. The messages in the corpus fall into four roughly equal-sized categories: Product Availability, Order Status, Order Cancellation, and Product Return.

**Meeting/hotel scheduling (Verbmobil)**  This dataset is a collection of transcriptions of spoken dialogues, each of which reflected a negotiation either to schedule a meeting, or to plan a hotel stay. One dialogue usually consists of 20-30 turns, with most of the utterances relatively short, including greetings and closings, and not surprisingly with a high frequency of time and date expressions as well as questions and sentence fragments.

**Norwegian tourism (LOGON)**  The Norwegian/English machine translation research project LOGON acquired for its development and evaluation corpus a set of tourism brochures originally written in Norwegian and then professionally translated into English. The corpus consists almost entirely of declarative sentences and many sentence fragments, where the average number of tokens per item is higher than in the Verbmobil and E-commerce data.

**The Tanaka Corpus (Tanaka)**  This treebank is based on parallel Japanese-English sentences, which was adopted to be used with in the WWWJDIC dictionary server as a set of example sentences associated within words in the dictionary.

## E GRAPH MATCHING ALGORITHM IN SMATCH

In general, finding the largest common subgraph is a well-known computationally intractable problem in graph theory. However, for graph parsing problems where graphs have labels and a simple tree-like structure, some efficient heuristics are proposed to approximate the best match by a hill-climbing algorithm (Cai & Knight, 2013). The initial match is modified iteratively to optimize the total number of matches with a predefined number of iterations (default value set to 5). This algorithm is very efficient and effective, it was also used to calculate the SMATCH score in Cai & Knight (2013).

## F  COMPARING T5 TO PREVIOUS WORK

Table 3 shows the in-domain performance of the T5 model compare to previous work on Redwoods, SMCalflow and SNIPS. The results indicate that the T5 model is capable of achieving the state-of-the-art results on all the tasks compared to previous work. Interestingly, we notice that by rewriting SMCalFlow into PENMAN notation, the sequence accuracy increase from 72.9 to 82.8 on development set, indicating that proper linearization and tokenization are important for graph parsing tasks.

| Redwoods (In-domain) | | | | SMCalflow | | SNIPS | |
|---|---|---|---|---|---|---|---|
| **Model** | **Node** | **Edge** | **SMATCH** | **Model** | **ACC$_{seq}$** | **Model** | **Slot F1** |
| Chen et al. (2018) | 94.5 | 87.3 | 90.9 | Andreas et al. (2020) | 72.9 | Wang et al. (2018) | 93.5 |
| Chen et al. (2019) | 97.3 | 94.0 | 95.7 | | | Zhang et al. (2019) | 91.8 |
| Cao et al. (2021) | 96.4 | 93.7 | 95.1 | | | Qin et al. (2019) | 94.2 |
| T5 (Lin et al., 2022b) | 97.3 | **95.8** | 96.6 | | | | |
| T5 (Our implementation) | **97.4** | 94.4 | **95.9** | | **82.8** | | **98.5** |

Table 3: In-domain performance evaluation for three semantic parsing tasks. The T5 model is built based on Lin et al. (2022b) on Redwoods and it surprisingly achieves the state-of-the-art results on the other datasets (SMCalflow and SNIPS).

## G  FULL RESULTS FOR COMPARING UNCERTAINTY BASELINES

In Table 4, we reported the full results for comparing different uncertainty baselines on the benchmark introduced in Section 3.

## H  HISTOGRAM OF CALIBRATIONS

The correlations between the subgraph's probability and performance on ERG parsing are shown in Figure 5, where we can see that low model probability generally corresponds to low model performance, i.e., the model is relatively calibrated in predicting graph structures.

## I  FULL RESULTS FOR COLLABORATIVE SEMANTIC PARSING

In Table 5 and Table 6, we report the full results for collaborative performance under different settings of budgets of subgraphs by different combinations of number of subgraphs ($e$) and max depth of each subgraphs ($d$). Note that for SNIPS, since the performance is almost close to 100% when selecting number of subgraphs greater than 3, i.e., the total graph has been covered, we will not evaluate cases where $e > 3$. We can see from the table that uncertainty-based subgraph selection consistently outperforms random subgraph selection, and performs close to oracle subgraph selection.

In Table 7, We further conduct an analysis to subgraphs selected by different strategies by calculating the coverage rate of incorrect nodes in subgraphs to incorrect nodes in the entire graph (i.e., error node coverage rate). The results indicates that compositional uncertainty is effective in detecting incorrectly predicted nodes.

## J  LIMITATIONS

Here we discuss some potential limitations of the current study:

**Linguistic Breadth**  The GAP model in this work is a general uncertainty quantification framework for graph parsing problems using seq2seq model, which theoretically has no restriction to formalism and languages adopted for the output graph. In this work, we have tested GAP on Redwoods, SMCalflow and SNIPS, which are all English based, but it is worth to see how the approach

| | | $\text{ACC}_{\text{seq}}(\uparrow)$ | $\text{SMATCH}(\uparrow)$ | $\text{F1}_{\mathbf{N}}(\uparrow)$ | $\text{F1}_{\mathbf{E}}(\uparrow)$ | $\text{ECE}_{\text{seq}}(\downarrow)$ | $\text{CECE}_G(\downarrow)$ | $\text{CECE}_{\mathbf{N}}(\downarrow)$ | $\text{CECE}_{\mathbf{E}}(\downarrow)$ |
|---|---|---|---|---|---|---|---|---|---|
| | Deterministic | 51.01 | 95.92 | 97.38 | 94.35 | 0.3632 | 0.0345 | 0.0221 | 0.0465 |
| | MC Dropout | 50.17 | 95.83 | 97.32 | 94.19 | 0.1034 | 0.0312 | 0.0171 | 0.0460 |
| | DE | 51.70 | 96.07 | 97.46 | 94.54 | 0.2545 | 0.0326 | 0.0202 | 0.0448 |
| WSJ (In-domain) | BE | 49.13 | 95.91 | 97.33 | 94.37 | 0.3209 | 0.0341 | 0.0217 | 0.0452 |
| | SNGP | 48.50 | 95.85 | 97.39 | 94.35 | 0.2213 | 0.0326 | 0.0199 | 0.0453 |
| | SNGP+DE | 49.55 | 95.98 | 97.45 | 94.53 | 0.1067 | 0.0299 | 0.0166 | 0.0431 |
| | SNGP+BE | 47.46 | 95.74 | 97.27 | 94.30 | 0.1652 | 0.0338 | 0.0202 | 0.0472 |
| | Deterministic | 68.10 | 95.27 | 95.65 | 94.95 | 0.2562 | 0.0452 | 0.0405 | 0.0498 |
| | MC Dropout | 67.53 | 95.20 | 95.55 | 94.90 | 0.1158 | 0.0405 | 0.0326 | 0.0492 |
| | DE | 69.42 | 95.45 | 95.79 | 95.17 | 0.1987 | 0.0411 | 0.0357 | 0.0467 |
| Tanaka | BE | 67.67 | 95.24 | 95.65 | 94.88 | 0.2367 | 0.0450 | 0.0395 | 0.0512 |
| | SNGP | 67.10 | 95.25 | 95.57 | 95.57 | 0.1998 | 0.0429 | 0.0367 | 0.0501 |
| | SNGP+DE | 68.03 | 95.51 | 95.83 | 95.30 | 0.1287 | 0.0394 | 0.0320 | 0.0477 |
| | SNGP+BE | 65.88 | 95.24 | 95.53 | 95.09 | 0.1913 | 0.0427 | 0.0363 | 0.0494 |
| | Deterministic | 51.53 | 93.48 | 94.18 | 92.88 | 0.3840 | 0.0721 | 0.0652 | 0.0793 |
| | MC Dropout | 49.46 | 93.04 | 93.72 | 92.50 | 0.1661 | 0.0725 | 0.0618 | 0.0835 |
| | DE | 53.14 | 93.72 | 94.38 | 93.15 | 0.3083 | 0.0676 | 0.0603 | 0.0754 |
| Ecommerce | BE | 50.63 | 93.66 | 94.29 | 93.14 | 0.3558 | 0.0677 | 0.0608 | 0.0740 |
| | SNGP | 50.43 | 93.09 | 93.79 | 93.79 | 0.2748 | 0.0743 | 0.0650 | 0.0849 |
| | SNGP+DE | 50.99 | 93.22 | 93.89 | 92.73 | 0.1848 | 0.0706 | 0.0599 | 0.0818 |
| | SNGP+BE | 49.28 | 93.17 | 93.91 | 92.65 | 0.2678 | 0.0739 | 0.0652 | 0.0832 |
| | Deterministic | 40.93 | 92.32 | 93.81 | 90.89 | 0.3969 | 0.0673 | 0.0530 | 0.0824 |
| | MC Dropout | 40.10 | 92.17 | 93.64 | 90.76 | 0.0982 | 0.0600 | 0.0409 | 0.0779 |
| | DE | 41.75 | 92.54 | 94.00 | 91.14 | 0.2589 | 0.0622 | 0.0473 | 0.0768 |
| Brown | BE | 40.19 | 92.07 | 93.69 | 90.54 | 0.3179 | 0.0690 | 0.0524 | 0.0854 |
| | SNGP | 36.98 | 91.93 | 93.44 | 90.76 | 0.2409 | 0.0666 | 0.0492 | 0.0832 |
| | SNGP+DE | 38.63 | 92.26 | 93.71 | 91.13 | 0.1119 | 0.0597 | 0.0417 | 0.0770 |
| | SNGP+BE | 36.94 | 91.83 | 93.39 | 90.63 | 0.1905 | 0.0655 | 0.0481 | 0.0823 |
| | Deterministic | 30.80 | 92.07 | 94.18 | 90.96 | 0.4589 | 0.0691 | 0.0565 | 0.0808 |
| | MC Dropout | 29.10 | 91.82 | 93.22 | 90.78 | 0.0993 | 0.0615 | 0.0464 | 0.0783 |
| | DE | 31.81 | 92.35 | 93.70 | 91.34 | 0.2859 | 0.0630 | 0.0505 | 0.0750 |
| Essay | BE | 27.75 | 91.71 | 93.22 | 90.54 | 0.3811 | 0.0725 | 0.0587 | 0.0855 |
| | SNGP | 25.04 | 91.61 | 93.16 | 90.96 | 0.2188 | 0.0675 | 0.0551 | 0.0822 |
| | SNGP+DE | 25.89 | 92.02 | 93.53 | 91.35 | 0.1124 | 0.0627 | 0.0474 | 0.0774 |
| | SNGP+BE | 24.03 | 91.62 | 93.15 | 91.01 | 0.2207 | 0.0693 | 0.0544 | 0.0840 |
| | Deterministic | 31.45 | 91.10 | 92.58 | 90.70 | 0.5148 | 0.0834 | 0.0726 | 0.0944 |
| | MC Dropout | 30.71 | 90.99 | 92.43 | 90.64 | 0.1457 | 0.0760 | 0.0628 | 0.0893 |
| | DE | 31.82 | 91.28 | 92.76 | 90.89 | 0.3393 | 0.0789 | 0.0672 | 0.0903 |
| LOGON | BE | 30.82 | 90.85 | 92.40 | 90.32 | 0.4288 | 0.0852 | 0.0738 | 0.0956 |
| | SNGP | 26.65 | 89.81 | 91.21 | 91.21 | 0.1553 | 0.0802 | 0.0667 | 0.0933 |
| | SNGP+DE | 26.60 | 90.75 | 92.24 | 90.86 | 0.0979 | 0.0752 | 0.0610 | 0.0890 |
| | SNGP+BE | 24.49 | 90.01 | 91.36 | 90.31 | 0.1555 | 0.0810 | 0.0695 | 0.0926 |
| | Deterministic | 51.45 | 90.38 | 91.45 | 90.02 | 0.3298 | 0.0960 | 0.0861 | 0.1063 |
| | MC Dropout | 52.74 | 90.42 | 91.54 | 89.92 | 0.1003 | 0.0873 | 0.0731 | 0.1033 |
| | DE | 54.56 | 90.67 | 91.69 | 90.35 | 0.2283 | 0.0909 | 0.0834 | 0.0996 |
| Verbmobil | BE | 51.56 | 90.00 | 91.12 | 89.58 | 0.2933 | 0.1010 | 0.0900 | 0.1126 |
| | SNGP | 52.63 | 90.21 | 91.30 | 91.30 | 0.2233 | 0.0981 | 0.0850 | 0.1102 |
| | SNGP+DE | 56.07 | 90.81 | 91.90 | 90.47 | 0.1196 | 0.0861 | 0.0737 | 0.0999 |
| | SNGP+BE | 53.92 | 90.42 | 91.48 | 89.99 | 0.2091 | 0.0943 | 0.0840 | 0.1060 |
| | Deterministic | 28.16 | 89.64 | 90.65 | 89.37 | 0.4105 | 0.1038 | 0.0939 | 0.1135 |
| | MC Dropout | 26.63 | 90.10 | 91.04 | 89.89 | 0.0492 | 0.0891 | 0.0731 | 0.1039 |
| | DE | 31.22 | 90.71 | 91.65 | 90.46 | 0.1974 | 0.0910 | 0.0798 | 0.1019 |
| Wiki | BE | 29.15 | 89.86 | 90.97 | 89.49 | 0.2658 | 0.0993 | 0.0881 | 0.1107 |
| | SNGP | 24.87 | 89.57 | 90.59 | 89.72 | 0.1631 | 0.0927 | 0.0804 | 0.1071 |
| | SNGP+DE | 27.31 | 90.63 | 91.58 | 90.69 | 0.0666 | 0.0803 | 0.0663 | 0.0942 |
| | SNGP+BE | 26.78 | 89.74 | 90.89 | 89.82 | 0.1281 | 0.0907 | 0.0777 | 0.1025 |

Table 4: Evaluations for different uncertainty baselines on different domains in Redwoods treebanks. BE refers to Batch Ensemble, and DE refers to Deep Ensemble.

can scale up (with the number of languages covered) and down (with diminished resource availability in low-resource languages). For example, many efforts have been made to build a practical and cross-linguistically valid graph formalism to overcome language barriers, e.g., Uniform Meaning Representation (UMR; Van Gysel et al., 2021) and BabelNet Meaning Representation (BMR; Navigli et al., 2022), and it is interesting to explore the model behavior on these graph formalism in terms of compositional uncertainty.

**Graphical Model Specification** The GAP model presented in this work considers a classical graphical model likelihood $p(G|x) = \prod_{v \in G} p(v|\operatorname{pa}(v), x)$, which leads to a clean factorization between graph elements $v$ and fast probability computation. However, it also assumes a local Markov property that $v$ is conditional independent to its ancestors given the parent $\operatorname{pa}(v)$. In theory, the

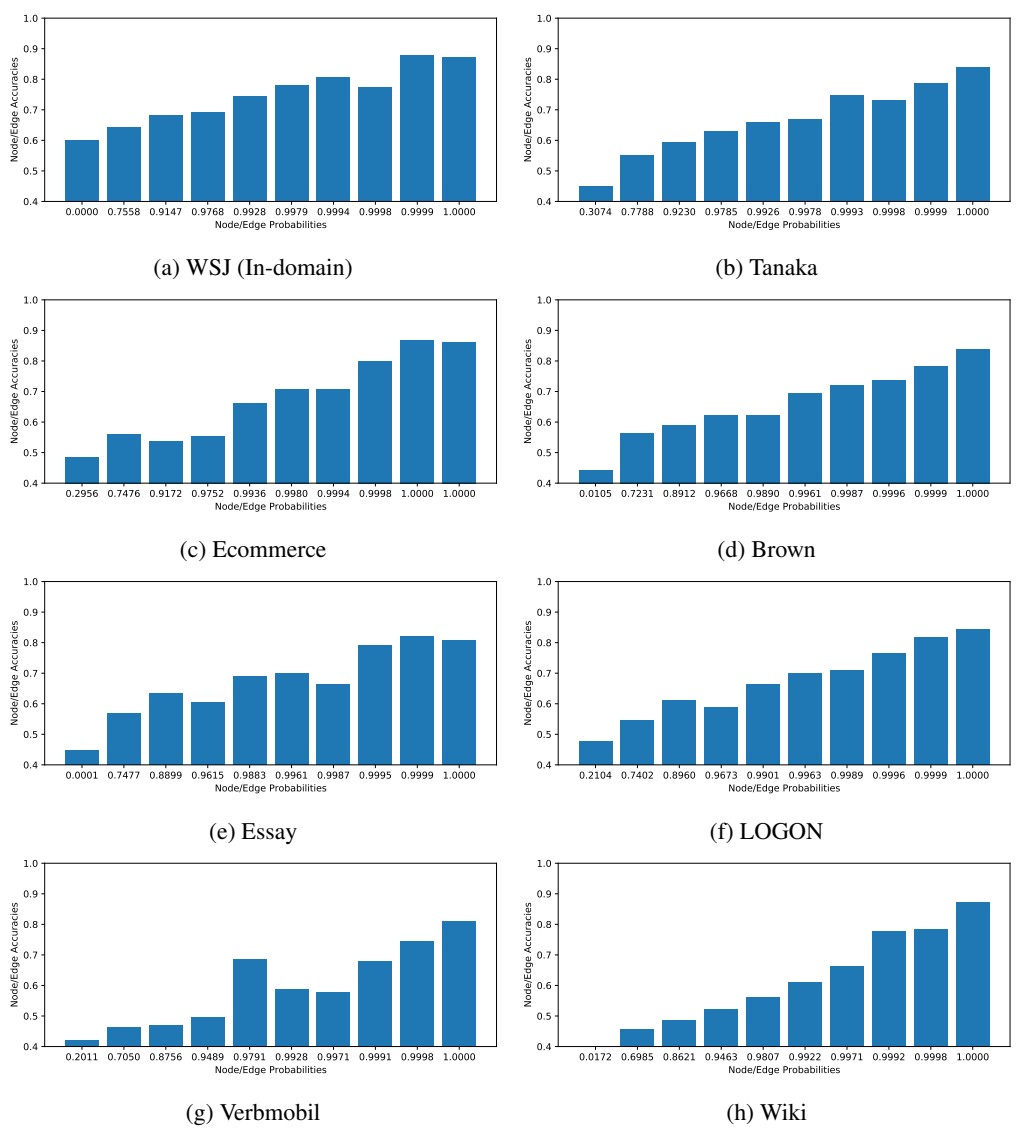

Figure 5: Diagrams for the T5 model's probabilities verses the T5 model's accuracies at subgraph level (nodes and edges). Each bin contains the same number of examples. Since at most of the subgraphs, the model is pretty certain ($\log P > -1e-5$), we exclude these pretty certain predictions in the figures.

probability learned by a seq2seq model is capable of modeling higher order conditionals between arbitrary elements on the graph. Therefore it is interesting to ask if a more sophisticated graphical model with higher-order dependency structure can lead to better performance in practice while maintaining reasonable computational complexity.

| Dataset | Retrival Type | Random | Det | MCD | GP | BE | BEGP | DE | DEGP | Oracle |
|---|---|---|---|---|---|---|---|---|---|---|
| WSJ (In-domain) | c=1,d=1 | 96.05 | 96.26 | 96.24 | 96.16 | 96.24 | 96.14 | 96.22 | 96.21 | 96.15 |
| | c=1,d=2 | 96.02 | 96.42 | 96.47 | 96.27 | 96.49 | 96.21 | 96.47 | 96.37 | 96.67 |
| | c=1,d=3 | 96.27 | 96.60 | 96.59 | 96.45 | 96.52 | 96.33 | 96.53 | 96.52 | 96.68 |
| | c=3,d=1 | 96.07 | 96.41 | 96.43 | 96.34 | 96.45 | 96.29 | 96.42 | 96.47 | 96.56 |
| | c=3,d=2 | 96.42 | 97.08 | 97.17 | 96.89 | 97.15 | 96.75 | 97.12 | 96.99 | 97.47 |
| | c=3,d=3 | 96.53 | 97.28 | 97.24 | 96.88 | 97.27 | 96.67 | 97.27 | 96.91 | 97.52 |
| | c=5,d=1 | 96.17 | 96.60 | 96.56 | 96.54 | 96.67 | 96.44 | 96.53 | 96.57 | 96.85 |
| | c=5,d=2 | 96.70 | 97.33 | 97.46 | 97.06 | 97.42 | 97.00 | 97.42 | 97.30 | 97.68 |
| | c=5,d=3 | 97.03 | 97.67 | 97.80 | 97.28 | 97.76 | 97.25 | 97.76 | 97.44 | 97.91 |
| Tanaka | c=1,d=1 | 95.35 | 96.23 | 96.18 | 96.02 | 96.24 | 96.01 | 96.21 | 96.14 | 96.22 |
| | c=1,d=2 | 95.88 | 96.72 | 96.75 | 96.45 | 96.63 | 96.31 | 96.69 | 96.54 | 96.92 |
| | c=1,d=3 | 96.03 | 96.96 | 97.00 | 96.60 | 96.91 | 96.52 | 96.94 | 96.70 | 97.27 |
| | c=3,d=1 | 96.15 | 96.79 | 96.94 | 96.77 | 96.84 | 96.85 | 96.88 | 96.91 | 97.11 |
| | c=3,d=2 | 96.96 | 97.83 | 97.94 | 97.65 | 97.86 | 97.73 | 97.91 | 97.85 | 98.04 |
| | c=3,d=3 | 97.15 | 98.18 | 98.43 | 98.31 | 98.43 | 98.19 | 98.32 | 98.36 | 98.43 |
| | c=5,d=1 | 96.57 | 97.25 | 97.38 | 97.30 | 97.26 | 97.37 | 97.27 | 97.37 | 97.53 |
| | c=5,d=2 | 97.40 | 98.38 | 98.49 | 98.34 | 98.41 | 98.47 | 98.43 | 98.55 | 98.44 |
| | c=5,d=3 | 97.99 | 98.97 | 99.09 | 98.94 | 98.99 | 99.05 | 99.09 | 99.13 | 98.89 |
| Ecommerce | c=1,d=1 | 93.86 | 94.54 | 94.47 | 94.14 | 94.95 | 94.23 | 94.76 | 94.29 | 94.98 |
| | c=1,d=2 | 94.49 | 95.06 | 94.87 | 94.58 | 94.97 | 94.45 | 94.45 | 94.76 | 95.83 |
| | c=1,d=3 | 94.75 | 95.49 | 95.58 | 94.95 | 95.56 | 94.83 | 95.61 | 95.14 | 96.16 |
| | c=3,d=1 | 94.43 | 95.47 | 95.63 | 95.21 | 95.75 | 95.30 | 95.68 | 95.50 | 95.91 |
| | c=3,d=2 | 95.02 | 96.31 | 96.68 | 96.40 | 96.54 | 96.31 | 96.66 | 96.35 | 97.18 |
| | c=3,d=3 | 96.22 | 96.62 | 97.04 | 96.76 | 97.00 | 96.59 | 97.04 | 96.76 | 97.61 |
| | c=5,d=1 | 95.21 | 95.83 | 95.90 | 95.80 | 95.99 | 95.88 | 95.98 | 95.98 | 96.28 |
| | c=5,d=2 | 96.30 | 97.20 | 97.61 | 97.24 | 97.55 | 97.09 | 97.47 | 97.31 | 97.67 |
| | c=5,d=3 | 96.83 | 97.85 | 98.12 | 97.97 | 98.30 | 98.03 | 98.13 | 98.15 | 98.45 |
| Brown | c=1,d=1 | 92.54 | 92.89 | 92.98 | 92.77 | 92.97 | 92.69 | 92.95 | 92.91 | 92.88 |
| | c=1,d=2 | 92.76 | 93.47 | 93.46 | 93.18 | 93.48 | 93.09 | 93.42 | 93.34 | 93.80 |
| | c=1,d=3 | 92.78 | 93.45 | 93.60 | 93.41 | 93.53 | 93.08 | 93.59 | 93.53 | 93.80 |
| | c=3,d=1 | 92.66 | 93.44 | 93.50 | 93.49 | 93.56 | 93.33 | 93.66 | 93.59 | 93.76 |
| | c=3,d=2 | 93.53 | 94.79 | 95.04 | 94.49 | 94.89 | 94.16 | 94.98 | 94.68 | 95.24 |
| | c=3,d=3 | 93.88 | 95.27 | 95.53 | 94.79 | 95.39 | 94.48 | 95.34 | 95.17 | 95.87 |
| | c=5,d=1 | 93.13 | 93.92 | 94.04 | 94.07 | 93.98 | 93.89 | 94.06 | 94.17 | 94.32 |
| | c=5,d=2 | 94.06 | 95.33 | 95.59 | 95.16 | 95.51 | 94.80 | 95.62 | 95.39 | 95.76 |
| | c=5,d=3 | 94.61 | 96.22 | 96.41 | 95.96 | 96.27 | 95.75 | 96.32 | 95.99 | 96.61 |
| Essay | c=1,d=1 | 92.38 | 92.62 | 92.73 | 92.82 | 92.70 | 92.56 | 92.77 | 92.63 | 92.57 |
| | c=1,d=2 | 92.15 | 92.83 | 92.70 | 92.32 | 92.61 | 92.35 | 92.66 | 92.64 | 92.99 |
| | c=1,d=3 | 92.36 | 92.92 | 92.92 | 92.63 | 92.99 | 92.56 | 93.02 | 92.96 | 93.10 |
| | c=3,d=1 | 92.30 | 93.29 | 93.15 | 93.15 | 93.30 | 92.97 | 93.11 | 93.28 | 93.40 |
| | c=3,d=2 | 92.82 | 94.21 | 94.03 | 93.76 | 93.82 | 93.48 | 94.10 | 93.75 | 94.82 |
| | c=3,d=3 | 93.13 | 94.64 | 94.76 | 94.06 | 94.14 | 93.96 | 94.44 | 94.39 | 95.38 |
| | c=5,d=1 | 92.26 | 93.26 | 93.25 | 93.30 | 93.52 | 93.27 | 93.55 | 93.65 | 93.44 |
| | c=5,d=2 | 93.66 | 94.84 | 94.88 | 94.36 | 94.91 | 94.16 | 94.79 | 94.48 | 95.13 |
| | c=5,d=3 | 93.99 | 95.68 | 95.47 | 95.38 | 95.77 | 94.96 | 95.75 | 95.23 | 96.55 |
| LOGON | c=1,d=1 | 91.42 | 91.93 | 91.75 | 91.60 | 91.78 | 91.61 | 91.86 | 91.70 | 91.72 |
| | c=1,d=2 | 91.85 | 92.37 | 92.41 | 92.05 | 92.47 | 91.96 | 92.53 | 92.11 | 92.75 |
| | c=1,d=3 | 91.87 | 92.52 | 92.45 | 91.93 | 92.54 | 91.88 | 92.53 | 92.03 | 92.83 |
| | c=3,d=1 | 91.60 | 92.49 | 92.56 | 92.56 | 92.49 | 92.38 | 92.66 | 92.50 | 92.96 |
| | c=3,d=2 | 92.65 | 93.68 | 93.74 | 93.09 | 93.74 | 93.19 | 93.74 | 93.33 | 94.38 |
| | c=3,d=3 | 92.95 | 94.19 | 94.40 | 93.34 | 94.23 | 93.19 | 94.15 | 93.60 | 95.10 |
| | c=5,d=1 | 91.92 | 92.77 | 92.97 | 92.92 | 92.78 | 92.81 | 92.81 | 92.77 | 93.28 |
| | c=5,d=2 | 93.42 | 94.61 | 94.75 | 94.00 | 94.58 | 93.85 | 94.45 | 94.22 | 95.22 |
| | c=5,d=3 | 93.76 | 95.24 | 95.40 | 94.51 | 95.32 | 94.31 | 95.29 | 94.93 | 95.77 |
| Verbmobil | c=1,d=1 | 90.16 | 90.69 | 90.77 | 90.59 | 90.76 | 90.43 | 90.74 | 90.68 | 90.76 |
| | c=1,d=2 | 90.63 | 91.92 | 91.51 | 91.44 | 91.65 | 91.08 | 91.84 | 91.59 | 91.76 |
| | c=1,d=3 | 90.78 | 91.92 | 91.76 | 91.83 | 92.10 | 91.66 | 91.88 | 91.86 | 91.68 |
| | c=3,d=1 | 89.98 | 91.42 | 91.67 | 91.68 | 91.72 | 91.35 | 91.59 | 91.65 | 91.14 |
| | c=3,d=2 | 92.15 | 93.75 | 94.13 | 94.04 | 94.03 | 93.54 | 94.00 | 93.93 | 94.18 |
| | c=3,d=3 | 93.36 | 95.32 | 95.44 | 95.07 | 95.15 | 94.68 | 95.24 | 95.20 | 95.40 |
| | c=5,d=1 | 90.98 | 92.74 | 92.78 | 92.70 | 92.89 | 92.80 | 92.49 | 92.86 | 92.37 |
| | c=5,d=2 | 94.00 | 95.35 | 95.47 | 95.38 | 95.33 | 94.85 | 95.29 | 95.39 | 95.43 |
| | c=5,d=3 | 94.56 | 96.45 | 96.59 | 96.53 | 96.53 | 96.04 | 96.50 | 96.88 | 96.26 |
| Wiki | c=1,d=1 | 91.34 | 91.82 | 91.78 | 91.73 | 91.82 | 91.70 | 91.75 | 91.72 | 91.84 |
| | c=1,d=2 | 91.88 | 92.14 | 92.34 | 91.98 | 92.30 | 92.09 | 92.25 | 91.98 | 92.70 |
| | c=1,d=3 | 91.74 | 92.22 | 92.32 | 92.00 | 92.53 | 91.77 | 92.35 | 91.96 | 92.53 |
| | c=3,d=1 | 91.48 | 92.15 | 92.18 | 92.37 | 92.20 | 92.34 | 92.23 | 92.40 | 92.38 |
| | c=3,d=2 | 92.07 | 93.08 | 93.17 | 92.75 | 93.12 | 92.63 | 93.23 | 92.92 | 93.49 |
| | c=3,d=3 | 92.17 | 93.20 | NaN | 93.14 | 93.33 | 93.02 | 93.58 | 93.17 | 93.84 |
| | c=5,d=1 | 91.66 | 92.64 | 92.66 | 92.58 | 92.80 | 92.56 | 92.66 | 92.87 | 92.76 |
| | c=5,d=2 | 92.92 | 93.97 | 94.00 | 93.80 | 94.14 | 93.59 | 94.24 | 93.80 | 94.67 |
| | c=5,d=3 | 93.07 | 94.43 | 94.51 | 93.89 | 94.44 | 93.68 | 94.79 | 94.07 | 95.18 |

Table 5: Full collaborative performance on Redwoods (ERG parsing) including different uncertainty baselines (Det: Deterministic; MCD: MC Dropout; GP: SNGP; BE: Batch Ensemble; BEGP: SNGP+Batch Ensemble; DE: Deep Ensemble; DEGP: SNGP+Deep Ensemble), the performance metrics are SMATCH score for Redwoods (Red: Best; Green: Second Best; Orange: Third).

| | | e=1_d=1 | e=1_d=2 | e=1_d=3 | e=3_d=1 | e=3_d=2 | e=3_d=3 | e=5_d=1 | e=5_d=2 | e=5_d=3 |
|---|---|---|---|---|---|---|---|---|---|---|
| SMCalflow | Oracle | 90.57 | 92.44 | 93.33 | 93.21 | 95.84 | 96.10 | 94.27 | 96.38 | 97.16 |
| | Oracle* | 91.01 | 93.38 | 94.55 | 93.41 | 96.56 | 97.36 | 94.19 | 97.39 | 97.87 |
| | Random | 88.10 | 88.99 | 89.71 | 90.10 | 92.21 | 93.47 | 91.30 | 93.80 | 95.19 |
| | Uncertain | 90.12 | 91.15 | 92.13 | 92.18 | 94.50 | 95.50 | 93.27 | 96.00 | 96.88 |
| SNIPS | Oracle | 99.48 | 99.54 | 99.56 | 99.70 | 99.74 | 99.78 | - | - | - |
| | Oracle* | 99.64 | 99.72 | 99.80 | 99.78 | 99.90 | 99.98 | - | - | - |
| | Random | 99.36 | 99.36 | 99.50 | 99.44 | 99.74 | 99.76 | - | - | - |
| | Uncertain | 99.62 | 99.62 | 99.70 | 99.82 | 99.90 | 99.98 | - | - | - |

Table 6: Full collaborative parsing performance on SMCalflow and SNIPS, the performance metrics are sequence accuracy for SMCalflow and slot (node) F1 for SNIPS.

| Dataset | Selection | e=1,d=1 | e=1,d=2 | e=1,d=3 | e=3,d=1 | e=3,d=2 | e=3,d=3 | e=5,d=1 | e=5,d=2 | e=5,d=3 |
|---|---|---|---|---|---|---|---|---|---|---|
| WSJ (In-domain) | Random | 3.65 | 7.21 | 8.79 | 9.19 | 19.33 | 23.78 | 14.29 | 28.69 | 33.14 |
| | Uncertain | 12.16 | 18.46 | 20.43 | 24.09 | 36.04 | 40.75 | 29.85 | 45.04 | 49.67 |
| | Oracle | 24.54 | 35.82 | 36.20 | 49.08 | 68.52 | 70.44 | 62.31 | 80.10 | 81.32 |
| Tanaka | Random | 11.78 | 20.19 | 22.05 | 29.33 | 46.81 | 48.00 | 39.13 | 58.20 | 64.05 |
| | Uncertain | 30.88 | 42.22 | 45.24 | 47.30 | 66.09 | 71.37 | 56.56 | 76.91 | 81.16 |
| | Oracle | 47.61 | 63.33 | 65.96 | 79.87 | 90.38 | 92.07 | 87.54 | 93.80 | 94.68 |
| Ecommerce | Random | 9.69 | 16.75 | 21.39 | 23.73 | 36.53 | 44.91 | 32.67 | 49.80 | 54.25 |
| | Uncertain | 23.75 | 33.17 | 34.52 | 42.65 | 55.84 | 59.35 | 51.10 | 69.12 | 72.65 |
| | Oracle | 41.80 | 54.27 | 58.72 | 73.70 | 87.34 | 88.17 | 85.19 | 93.52 | 94.78 |
| Brown | Random | 5.38 | 11.06 | 12.47 | 14.90 | 27.47 | 31.13 | 21.70 | 37.98 | 42.26 |
| | Uncertain | 14.09 | 22.34 | 25.54 | 28.52 | 44.08 | 49.21 | 35.73 | 55.01 | 60.62 |
| | Oracle | 24.88 | 36.01 | 38.90 | 52.53 | 67.79 | 72.42 | 64.86 | 79.60 | 82.79 |
| Essay | Random | 4.88 | 8.40 | 12.92 | 11.51 | 25.74 | 26.16 | 18.96 | 35.82 | 41.22 |
| | Uncertain | 12.60 | 19.50 | 22.15 | 25.59 | 41.85 | 45.32 | 32.89 | 52.59 | 59.30 |
| | Oracle | 20.62 | 29.87 | 33.65 | 45.80 | 64.54 | 70.77 | 62.43 | 80.33 | 84.33 |
| LOGON | Random | 6.19 | 12.42 | 15.54 | 14.97 | 30.42 | 34.67 | 23.51 | 43.91 | 46.74 |
| | Uncertain | 14.68 | 23.15 | 25.49 | 29.95 | 46.00 | 50.49 | 38.34 | 58.14 | 64.29 |
| | Oracle | 25.50 | 37.31 | 42.11 | 56.44 | 74.25 | 74.57 | 69.88 | 85.82 | 87.71 |
| Verbmobil | Random | 12.49 | 23.80 | 25.68 | 27.55 | 43.78 | 48.09 | 39.97 | 57.85 | 62.55 |
| | Uncertain | 23.54 | 35.75 | 39.59 | 45.21 | 63.38 | 69.61 | 55.11 | 75.06 | 80.65 |
| | Oracle | 34.05 | 52.33 | 56.65 | 70.01 | 82.70 | 86.63 | 81.45 | 92.10 | 92.45 |
| Wiki | Random | 4.61 | 11.98 | 13.22 | 13.63 | 25.92 | 30.83 | 20.67 | 37.01 | 43.48 |
| | Uncertain | 14.18 | 21.61 | 23.13 | 28.64 | 42.78 | 47.16 | 36.62 | 53.76 | 57.82 |
| | Oracle | 22.13 | 32.57 | 33.78 | 47.54 | 62.21 | 66.99 | 59.19 | 76.53 | 79.26 |
| SMCalflow | Random | 10.90 | 16.24 | 21.28 | 21.27 | 33.55 | 44.07 | 29.30 | 45.24 | 56.20 |
| | Uncertain | 14.45 | 21.46 | 27.32 | 24.45 | 37.36 | 47.85 | 31.77 | 48.11 | 59.67 |
| | Oracle | 21.75 | 30.02 | 37.65 | 42.97 | 59.87 | 65.89 | 55.28 | 70.99 | 77.35 |
| SNIPS | Random | 39.02 | 67.07 | 56.10 | 65.85 | 86.59 | 90.24 | - | - | - |
| | Uncertain | 70.73 | 76.83 | 76.83 | 98.78 | 100.00 | 100.00 | - | - | - |
| | Oracle | 75.61 | 81.71 | 93.90 | 100.00 | 100.00 | 100.00 | - | - | - |

Table 7: Error node coverage rate for different subgraph selection strategies and budget ($e$ refers to number of examples (subgraphs), $d$ refers to the depth of the subgraphs).

