# OpenReview forum: "On Compositional Uncertainty Quantification for Seq2seq Graph Parsing"
_ICLR.cc/2023/Conference — ICLR 2023 poster_

### Official Review · Reviewer_oThz · 2022-10-23

**Confidence:** 3
**Correctness:** 4
**Technical Novelty And Significance:** 4
**Empirical Novelty And Significance:** 4
**Recommendation:** 8

**Clarity, Quality, Novelty And Reproducibility:**

First sentence of the intro, typo, graph paring -> graph parsing
Table 1 and fig 3 are extremely small

I find the paper technically sound and clear.

**Strength And Weaknesses:**

The paper is well written and seems to be a missing piece in the semantic parsing literature. Calibration error is a standard metric to understand how trustworthy the predictions of a model are, and apparently this is the first attempt at calculating it in the setting of compositional sequence to sequence models.

The paper is pretty clear and goes step by step explaining the various concepts used in the paper. it is equation-heavy but easy to understand.

The premises for section 3.2 are a bit vague, to me modelling compositional uncertainty would be very helpful in an active learning framework, where subgraph on which the model is uncertain are asked to be reannotated. The collaborative semantic parsing scenario is not very realistic to me.
It is not clear to me also how the oracle is defined.

**Summary Of The Paper:**

The paper propose a method to calculate the compositional uncertainty of sequence to sequence model for semantic parsing tasks, that is in tasks where the input is an utterance and the output is a graph.
For doing so the authors propose to marginalize the probability of output sequences so to calculate the expected probability of a node, give its parent node in a generated graph. Since marginalizing over all possible subsequences between two symbols is intractable in general, the authors propose to approximate this process using beam decoding.
The authors use the compositional uncertainty to calculate the expected calibration error, that is how well the uncertainty matches the performances of the model. The authors compare standard ECE and compositional ECE, showing that the second one is a better choice.

Finally the model is used in a collaborative semantic parsing where parts of the predictions (for example where the models is more uncertain on) can be send to human validation. Results show that the proposed measure of uncertainty finds more errors than random, and it is close to an oracle selection.

**Summary Of The Review:**

This work introduce a technique that allows practitioner to calculate compositional uncertainty in seq 2 seq model for graph parsing.
I am not an expert in graph parsing but it seems that this paper fills a void in that space.
I am not able to judge if more experiments on different datasets are required to prove if the approach is general enough to be used in any kind of text 2 graph task.

---

> ### Author Response · Authors · 2022-11-16
> **Response to Reviewer oThz**
>
> Dear Reviewer oThz,
>
> We really appreciate your comments and your support for our work. We hope our response can address your concerns.
>
> - In this work, we have focused on establishing the fundamental methodology for uncertainty quantification, and its rigorous evaluation. Therefore, for extrinsic evaluation, we have selected collaborative parsing that is of practical importance (i.e., can be directly applied to applications like clarification asking and dialog repair), clearly connected with uncertainty performance (i.e., compositional calibration), and currently underexplored in the literature. This methodology is also promising in many other applications, e.g., retrieval-based data augmentation, neural-symbolic parsing and active learning. In supplementary material, we have also illustrated the practical effectiveness of this approach in neural-symbolic parsing. However, building specific applications is not the focus of this paper and has been left for future work (see Section 4 for future directions).
> - *Oracle* is defined by sending the incorrectly predicted subgraphs for human review. We have reorganized Section 3.2 to improve the clarity.
> - In the updated paper, we have adjusted the size of Table 1 and Figure 3 to make it more clear.

---

### Official Review · Reviewer_jzBV · 2022-10-24

**Confidence:** 4
**Clarity, Quality, Novelty And Reproducibility:** 1. Probably should mention Vinyals et…
**Correctness:** 2
**Technical Novelty And Significance:** 2
**Empirical Novelty And Significance:** 3
**Recommendation:** 6

**Strength And Weaknesses:**

# Strengths

1. The paper is focused on seq2seq parsing with T5. This is a challenging task and active research area. They evaluate on many semantic parsing tasks.

2. The authors explicitly model the graph rather than solely treat it as a string-to-string task.

3. The authors evaluate in a creative scenario where humans can edit noisy graphs to drastically improve performance, although this section is limited.

# Weaknesses

1. There are many missing related work, and it is not right to say they are first to explore local probabilities for seq2seq parsing.

2. The evaluation is very hard to interpret. In the main table 1, there are no baselines and only T5 w/ GAP is evaluated as far as I can tell. In figure 3 there are many methods but I am not sure how it ties into the main storyline regarding T5 w/ GAP. In addition the evaluation is not very standard, which alone is not a major problem, but warrants further explanation in the text.

3. To me the most exciting part of the work is that noisy graphs can potentially be edited to improve performance. This section leads to some concerns... I am surprised RANDOM can be so effective. There are not really any details on how human edits are done. There is not much analysis on the subgraph editing, and it seems like most of editing can be done by editing one or two labels.

**Summary Of The Paper:**

The authors do seq2seq parsing with various semantic parsing datasets, and employ a specialized method called GAP that is meant to incorporate "compositional uncertainty" (UC). My interpretation is that UC incorporates probabilities over graph construction (i.e. edge and node creation) rather than token predictions when training seq2seq models with max-likelihood objectives (although there is a fine line between these two, and perhaps the former is achieved simply by using the right action space). Also, their training involves approximate normalization over subgraphs via importance sampling. Because of uncertainty measures that GAP implements, it is suitable for human-in-the-loop graph editing (sec 3.2). This last idea is quite interesting --- maybe your initial graph is fine and a human can slightly fix it up to drastically improve performance. That being said, the eval is somewhat limited and the main text is lacking details.

**Summary Of The Review:**

The core idea of the paper, that uncertainty can be used to inform graph editing, seems very interesting but I am not sure it is well executed. Parts of the presentation are confusing, and there could be more analysis on the graph editing portion. By fixing up writing, adding appropriate baselines, and providing more emphasis on graph editing part, then I think this could be a stronger paper. Right now I do not think it meets standards of the conference.

---

> ### Author Response · Authors · 2022-11-16
> **Response to Reviewer jzBV (3/3)**
>
> 10. Regarding editing subgraphs
> - There are three subgraph selection strategies given a certain budget (limited number and size of the subgraphs). For example, if we are allowed to send only one subgraph with a depth up to two for human review for each example (e=1; d=2), we have the following options:
>     - **Random**: We pick one random node and search depth up to two to form the subgraph.
>     - **Uncertain**: We pick the most uncertain node according to compositional uncertainty and search depth up to two to form the subgraph.
>     - **Oracle**: We pick the node which is incorrectly predicted by the model and search depth up to two to form the subgraph.
> - Regarding if subgraph editing is sufficient, our focus in this work is to evaluate the effectiveness of the proposed uncertainty quantification procedure, rather than proposing a complete solution to the graph editing problem.  Indeed, there will exist situations where fixing local graphs alone is not sufficient to obtain the fully correct solution (e.g., the predicted parent-child relations between subgraphs are wrong). However, even if so, an effective uncertainty quantification procedure can still select (and consequently correct) the most problematic subgraphs, and as a result can more effectively improve the SMATCH performance of the edited graph.
> - Probability as a notion of ambiguity: the probability $p(s|pa(s), x)$ reflects the model's degree ambiguity among plausible choices of the prediction candidate of the graph element s. Therefore, when there are many candidates, lower probability of $p(s|pa(s), x)$ indeed implies a more diffuse, closer-to-uniform predictive distribution among candidates. Furthermore, in the context of practical seq2seq inference, computing the predictive probability tends to be computationally more efficient than alternative notion of distribution ambiguity (e.g., entropy), as those usually require enumerating over a large space of all possible candidates.
> - Indicator for OOD: Indeed, when out-of-domain, the model tends to be less confident due to encountering novel noun phrases or linguistic phenomena. As a result, the proposed uncertainty procedure can become conversative when going out-of-domain. However, this in fact can be desirable as the neural models are known to tend to make mistakes in unfamiliar situations. Indeed, as our out-of-domain result shows (Table 5), our procedure is in fact empirically effective in the out-of-domain cases.
> 11. Regarding GAP formulation,
> - Thanks for this question. In most of the cases we consider (parsing trees and DAGs), the linearized sequence usually follows a pre-defined traversal order (e.g., BFS or DFS on tree or DAG, with a pre-specified ordering between child nodes), therefore this ambiguity between sequence to graph mapping does not frequently happen.
> - However, when the sequence-to-graph ambiguity does happen, GAP is able to naturally resolve this sequence-to-graph ambiguity by design, since it computes the $p(s|pa(s), x)$ by aggregating the probabilities for graph elements across multiple beam candidates.
> - As a result, the probability of multiple sequences corresponding to the same graph will be aggregated together to express the model's posterior belief for the same graph.
> - For example,  for two beam candidates $A \rightarrow B$ and $B \leftarrow A$ that express the same graph G with root $A$, directed edge $\rightarrow$ and child $B$. Then, given input $x$, we can compute the root probability $P(A|x)$ by aggregating probability from the two sequences as Equation (4), and compute the conditional probabilities $P(B|pa(B), x)$ by aggregating over the two sequences as Equation (5) (and apply Bayes rule as appropriate, as mentioned at the end of the paragraph below equation (3)). Consequently, GAP can naturally resolve this sequence-to-graph ambiguity by aggregating the statistical evidence for graph G over multiple sequences.
> 12. Thank you for providing these references. We have included this in our future work in Section 4(3).

---

> > ### Comment · Reviewer_jzBV · 2022-11-24
> > **Thanks**
> >
> > I appreciate the detailed response by the authors, which helps clarifies various points. I'm updating my score to 6, just above acceptance threshold.
> >
> > I still have some lingering concerns that would make this paper more approachable, although I'm not sure they're easily addressable in the reviewing period and I don't expect to improve my score. For instance:
> >
> > * The distinction between seq2seq parsing and transition-based parsing is shaky. I would recommend reframing the benefit of your approach without needing to draw this line. My impression is that seq2seq parsing (like Vinyals et al) is desirable because you can basically ignore graph structure, which is clearly not the case here. (related to clarity comments)
> >
> > * Minor note, but given the emphasis on compositionality, I think compound divergence is quite relevant here (see Keysers et al 2019). (related to weakness 1, although this weakness was substantially addressed with the response + revisions)
> >
> > * It's still not clear if CECE would work as well w/ models besides t5-large. Not a big deal, but does hurt your claim about this being a general approach. (weakness 2)
> >
> > * It's a bit deceptive to call these human edits, when they're simply copied from the ground truth. (related to weakness 3, although this became more clear with the revisions)
> >
> > Even with the concerns, I think the work would be interesting for many people at this time given the topical relevance of compositional uncertainty and the novel approach.
> >
> > Keyser et al 2019. Measuring Compositional Generalization: A Comprehensive Method on Realistic Data. https://openreview.net/forum?id=SygcCnNKwr

---

> > > ### Author Response · Authors · 2022-12-11
> > > **Re: Thanks**
> > >
> > > Dear Reviewer jzBV,
> > >
> > > Thanks for your response and comments! Regarding your concerns:
> > > - Our work can also be adapted to transition-based parsing but an extra process of transforming probability from action sequence to graph structure is required, and this usually means that the dataset has explicit node-to-word alignments (which is not the case for SMCalflow). To make this work model- and task-agnostic, we did not consider transition-based parsing, but it is worth trying in future work.
> > > - Thanks for mentioning this. On a high level, the compositional ECE (i.e., CECE) we proposed in this paper can be understood as a weighted combination of component-wise divergence metrics, where we selected ECE as the divergence metric in this work due to its widespread usage in the uncertainty quantification literature. Indeed, as a future direction, it is interesting to generalize CECE to a wider family of divergence metrics to incorporate different metric needs. This can include the Chernoff coefficient as proposed Keyser et al 2019, or the [strictly proper scoring rules](https://sites.stat.washington.edu/raftery/Research/PDF/Gneiting2007jasa.pdf) that is commonly adopted in the probabilistic forecast literature.
> > > - CECE is compatible with any seq2seq model as long as the autoregressive token-level probabilities are provided. As for collaborative parsing, a good performance relies on a relatively well-calibrated model on predicting graph structures, which can be validated by CECE.
> > > - Thanks for the comments, we will change it to oracle edits in the final version. Oracle edits are the first step before moving to more realistic human-in-the-loop applications, which can empirically test the effectiveness of utilizing compositional uncertainty without heavy human effort. In fact, many previous works have adopted a similar oracle setting, e.g., [Kivlichan et al., 2021](https://arxiv.org/abs/2107.04212) and [Stengel-Eskin et al., 2022](https://arxiv.org/abs/2211.07443).

---

> ### Author Response · Authors · 2022-11-16
> **Response to Reviewer jzBV (2/3)**
>
> 3. **Graph editing part (analysis and details)**
> - Regarding the performance of random subgraph selection, the performance of collaborative parsing is determined by how many incorrectly predicted subgraphs can be selected for human edits, where oracle is the headroom given a limited budget. In the updated draft, we add analysis for the coverage rate of incorrect nodes in subgraphs to incorrect nodes in the entire graph (i.e., error node coverage) to further illustrate the results in Figure 4 as follows (this table is based on WSJ in-domain, and the full table can be found in Table 7). As can be seen from the table, the random selection can cover a small number of incorrect subgraphs and lead to performance improvement, while uncertain subgraphs can cover much more subgraphs and lead to significant performance improvement. This indicates that **compositional uncertainty is effective in detecting model errors on subgraphs**.
>
> | **Subgraph Selection** | **e=1,d=2** | **e=3,d=2** | **e=5,d=2** |
> |------------------------|-------------|-------------|-------------|
> | Random                 | 7.21        | 9.19        | 28.69       |
> | Uncertain              | 18.46       | 36.04       | 45.04       |
> | Oracle                 | 35.82       | 68.52       | 81.32       |
>
> - Since we do not have access to expert annotators for these tasks, we run a simulated human-in-the-loop experiment by simulating an oracle annotator who always provides a correct answer by using the gold annotations provided in the datasets, i.e., human edits are done by making the model access to the corresponding gold subgraphs to the selected subgraphs.
>
> ### Clarity, Quality, Novelty And Reproducibility
> 1. Add this to the introduction section.
> 2. Add this to related work in Appendix A.
> 3. Thank you for pointing out these related works. In our initial version, some papers specific to the PENMAN format are not discussed in detail due to the space limit. In the updated draft, we have added them to Appendix A, where we also add those uncertainty related works.
> 4. Thanks for pointing this out. Indeed, the GAP model presented in this work assumes a local Markov property that $v$ is conditional independent to its ancestors given the parent $pa(v)$. In theory, the probability learned by a seq2seq model is capable of modeling higher-order conditionals between arbitrary elements on the graph. Therefore it is interesting to ask if a more sophisticated graphical model with a higher order dependency structure can lead to better performance in practice while maintaining reasonable computational complexity. We have added this discussion in Appendix J.
> 5. Please see the first point in the Strengths and Weaknesses.
> 6. Your understanding is correct, and long-range distance refers to sequence space. We have updated the statement to make it more clear.
> 7. Global likelihood is $p(s_i | x)$. It can be called cumulative likelihood or marginal likelihood (Equation 4). For a particular graph element $s_i,$ It is computed by cumulating over the space of graphs from the model's posterior distribution that contains $s_i$.
> 8. The statement "*sequence accuracy does not necessarily correlate to the Smatch score*" is not the conclusion but one of the reasons why Compositional ECE is better than vanilla ECE – it considers fine-grained structure prediction.
> 9. We chose PENMAN to be our linearization for graphs since it is effective in representing semantic graphs and has been widely used in graph-based meaning representation parsing such as AMR parsing. However, for seq2seq graph parsing, the variables in PENMAN format will confuse the model to learn the real meaningful mappings. Therefore, we remove the identifiers and use star markers instead to indicate reentrancies (more details can be found in updated Appendix B). This variable-free PENMAN notation has been shown effective in the previous seq2seq graph parsing work ([Lin et al., 2022](https://aclanthology.org/2022.findings-acl.328/)) and has the same effect as in [Bevilacqua et al. 2021](https://ojs.aaai.org/index.php/AAAI/article/view/17489).

---

> ### Author Response · Authors · 2022-11-16
> **Response to Reviewer jzBV (1/3)**
>
> Dear Reviewer jzBV,
>
> We really appreciate your comments. As you suggested, we have fixed up the writing, added appropriate baselines, and provided more analysis on collaborative parsing in the updated paper.  We hope our following point-to-point response can address your concerns and clarify our contribution.
>
> ### Strengths and Weaknesses
>
> 1. **Missing related work and clarification for contributions.**
> - Thank you for pointing out some related works, and we have updated the draft accordingly.
> - In the updated draft, we have made this claim more precise by saying that we are the first to provide a general probabilistic framework (GAP) that can quantify compositional uncertainty for seq2seq graph parsing. GAP allows us to go beyond the conventional sequence-level probability and express parent-child conditional probability on the graph, which is compatible with any graph parsing problems and autoregressive models.
> - We also want to highlight that in this work, seq2seq graph parsing refers to **directly predicts a linearized target graph without intermediate steps** (unlike other approaches such as chart parsers and transition-based parsers), and our goal is to **build a general uncertainty quantification framework for all seq2seq graph parsing problems**. See Problem Formulation in Section 2 for the detailed definition. Compositional uncertainty quantification for this type of parsing is challenging, important and underexplored:
>     - **It is different from uncertainty in chart parsers** ([Magernan & Marcus, 1991](https://aclanthology.org/1991.iwpt-1.22/)), factorization-based parsers ([McDonald, 2006](https://ryanmcd.github.io/papers/thesis.pdf); [Cao et al., 2021](https://aclanthology.org/2021.cl-1.3/)), and composition-based parsers ([Chen et al., 2018](https://aclanthology.org/P18-1038/); [Chen et al., 2019](https://aclanthology.org/K19-2016/)), since their score functions are naturally aligned with the graph structures. Here we need to address the gap between sequence-level probabilities and graphical probabilities.
>     - **It is also different from transition-based parsers** ([Drozdov et al. 2022](https://aclanthology.org/2022.naacl-main.80/)), where the target graph is generated via a series of actions based on rules and alignments. Compositional uncertainty quantification in our paper is more difficult than simply modeling the ambiguity of alignments between tokens and actions.
>     - **The parsers mentioned above are difficult to be transferred from one graph formalism to another** without checking the reusability of designed rules and model architectures, which is against our goal.
>     - **This approach (i.e., translation-based parsing) is currently the state-of-the-art approach for most NLP parsing applications** (see Appendix A.1 for details), which also has the benefit of being **task- and grammar-agnostic**, which is in accordance with our goal of producing a general-purpose uncertainty quantification method for graph-structured prediction problems. However, **there is no previous work** addressing compositional uncertainty quantification for seq2seq graph parsing problems.
>
> 2. **Clarification for evaluations**
> - Regarding Table 1, the point we would like to highlight here is that compared to vanilla ECE, compositional ECE is a better metric for reflecting distributional shift (i.e., domain distance or task ambiguity). In the updated draft, we have added a heated color background to illustrate this visually. This evaluation can be done by any seq2seq model and we take T5 as our case study here. The performance metrics guarantee that we are conducting evaluation based on a competitive seq2seq model. In the updated paper, we have added a full table (Table 3) to compare to previous work for these tasks (in-domain). **The results show that T5 achieves state-of-the-art results on all tasks compared to previous work**.
> - Regarding Figure 3, our goal here is to explore whether those uncertainty methods that have shown better uncertainty quality for classification tasks, still work better for compositional uncertainty in graph parsing. **The results reveal a non-obvious fact that the absolute advantage for those uncertainty methods holds for sequence ECE but not for compositional ECE**. This indicates that uncertainty estimation is structurally different for seq2seq graph parsing tasks compared to classification tasks, and further research is needed for designing better calibrated model with more focus on compositional uncertainty calibration, which cannot be achieved by properly quantifying compositional uncertainty.
> - In the updated paper, we have reorganized Section 3.2 to further explain the evaluation settings.

---

### Official Review · Reviewer_3RNa · 2022-10-27

**Confidence:** 3
**Correctness:** 4
**Technical Novelty And Significance:** 4
**Empirical Novelty And Significance:** 4
**Recommendation:** 10

**Clarity, Quality, Novelty And Reproducibility:**

The work is of exceptional quality and clarity, and it is both original and important to the (sub-)field.

**Details Of Ethics Concerns:**

None.

**Strength And Weaknesses:**

Strengths:
- A novel and creative contribution to modelling and evaluation for uncertainty in seq2seq parsing.
- Thorough experiments, high-quality writeup, convincing results.

Weaknesses:
- The selection of datasets is not thoroughly elaborated from the viewpoint of availability vs. linguistic breadth. English is dominant in graph parsing, but care should be taken to qualify how the approach would scale up (with the number of languages covered) and down (with diminished resource availability in low-resource languages). A Limitations section addressing this linguistic breadth angle, and an honest account of this contribution being by and large for English only, would suffice.

**Summary Of The Paper:**

The paper is a novel and profound contribution to sequence-to-sequence graph parsing of natural language, quantifying and evaluating compositional uncertainty in seq2seq graph parsing through a newly-proposed probabilistically interpretable framework and a new metric which measures a model's calibration in predicting graph structures. A large battery of tests together with human evaluations shows strong empirical support for the framework.

**Summary Of The Review:**

Contained above.

---

> ### Author Response · Authors · 2022-11-16
> **Response to Reviewer 3RNa**
>
> Dear Reviewer 3RNa,
>
> We really appreciate your comments and your support for our work. We hope our response can address your concerns.
>
> Regarding linguistic breadth, our goal in this paper is to provide a general uncertainty quantification framework for seq2seq graph parsing problems, which has no restriction to models or graph formalism, and also to languages. In fact, many efforts have been made to build a practical and cross-linguistically valid graph formalism to overcome language barriers, e.g., Uniform Meaning Representation ([UMR; Gysel et al., 2021](https://link.springer.com/article/10.1007/s13218-021-00722-w)) and BabelNet Meaning Representation ([BMR;  Navigli et al., 2022](https://ojs.aaai.org/index.php/AAAI/article/view/21490)), but these works about language resource construction are not within the scope of this paper regarding uncertainty quantification. As you suggested, in the updated draft, we have added a limitation section in Appendix J to address this linguistic breadth angle.

---

### Author Response · Authors · 2022-11-16
**General response to all reviewers and the new revision**

We sincerely thank all the reviewers for their feedback and constructive comments. We are pleased that the reviewers found our work exciting/creative (3RNa & jzBV) and novel (3RNa & oThz), the experimental results strong (3RNa), and the paper well-written (3RNa & oThz). Here we would like to highlight some points which the reviewers are most concerned about:

- **Clarification for our contributions:** Our goal is to propose a simple and general probabilistic framework (GAP) that can quantify compositional uncertainty for seq2seq graph parsing. The seq2seq parsing here directly predicts a linearized target graph without any intermediate step (i.e., unlike chart parsers or transition-based parsers), so GAP is compatible with any graph parsing problem (task/language agnostic) and autoregressive model (model agnostic). Compositional uncertainty quantification for seq2seq graph parsing is challenging, important, and underexplored, and we are the first to establish the fundamental methodology for this direction.

- **Collaborative parsing as extrinsic evaluation:** In this work, we have focused on establishing the fundamental methodology for compositional uncertainty quantification, and its rigorous evaluation. Therefore, for extrinsic evaluation, we have selected collaborative parsing that is of practical importance (i.e., can be directly applied to applications like clarification asking and dialog repair), clearly connected with uncertainty performance (i.e., compositional calibration), and currently underexplored in the literature. This methodology is also promising in many other applications, e.g., retrieval-based data augmentation, neural-symbolic parsing and active learning. In supplementary material, we have also illustrated the practical effectiveness of this approach in neural-symbolic parsing. However, building specific applications is not the focus of this paper and has been left for future work (see Section 4 for future directions).

We have revised and updated the draft (highlighted in blue color in the new pdf), which reflects the reviewers’ comments. The major revisions are:
- **Section 1:** We have updated the first point of our contributions to make it more precise (Reviewer jzBV).
- **Table 1:** We have added heated color to better highlight the advantage of compositional ECE (Reviewer jzBV).
- **Section 3.2:** We have reorganized this section and added an analysis (Table 7) (Reviewer jzBV & oThz).
- **Appendix A:** We have added more related work for graph parsing (Reviewer jzBV).
- **Appendix F:** We have added appropriate baselines (Reviewer jzBV).
- **Appendix J:** We add a discussion of the limitations of our work including linguistic breadth and local compositionality (Reviewer 3RNa & jzBV).

---

### Decision · Program_Chairs · 2023-01-20

**Decision:**

Accept: poster

**Justification For Why Not Higher Score:**

The paper makes a strong contribution to the problem of quantifying uncertainty in sequential models of structured prediction. While this could have a wider impact than just semantic graph parsing, the paper's audience will likely be more limited and therefore is most suited as a poster presentation.

**Justification For Why Not Lower Score:**

The paper makes novel contributions to the problem of quantifying uncertainty in sequential models for graph parsing, which is an important problem due to the challenges in approaching structured prediction as a sequential prediction problem.

**Metareview: Summary, Strengths And Weaknesses:**

The paper studies uncertainty quantification for sequence-to-sequence semantic graph parsers. The paper proposes a framework for mapping the probability of elements in the graph linearization to graph elements, together with a beam search approach to approximate marginalizing over subsequences. Compositional uncertainty is then quantified by a proposed compositional expected calibration error (CECE), which is compared to expected calibration error (ECE). Evaluation on 3 semantic graph parsing tasks and 10 domains shows that CECE correlates better with out of domain generalization than ECE under various uncertainty quantification methods. The results show structural differences between uncertainty in graph parsing compared to classification. The method is then applied as a proof of concept for collaborative semantic parsing, where performance can be improved by replacing a small number of uncertain subgraphs through human intervention (simulated here through an oracle). The proposed approach is a valuable contribution to the semantic parsing literature in terms of measuring a model’s calibration in predicting graph structures. Evaluating on more kinds of graph structures would be useful, and the experiments on graph editing could be extended, but the experiments are strong enough to support to claims of the paper and the overall contribution is strong.

**Note From Pc:**

if the above contains the word "oral" or "spotlight" please see: "oral" presentation means -> notable-top-5% and "spotlight" means -> notable-top-25%. As stated in our emails, we are disassociating presentation type from AC recommendations